# Perturbation Towards Easy Samples Improves Targeted Adversarial Transferability

**Junqi Gao**[*,1], **Biqing Qi**[*,2,3,4], **Yao Li**[†, 1], **Zhichang Guo**[1], **Dong Li**[1], **Yuming Xing**[1], **Dazhi Zhang**[1]

[1]School of Mathematics, Harbin Institute of Technology
[2]Department of Control Science and Engineering, Harbin Institute of Technology
[3]C[3]I, Department of Electronic Engineering, Tsinghua University
[4]Frontis.AI
∗ Equal contribution, † Corresponding author.
{gjunqi97, qibiqing7, mathgzc, arvinlee826}@gmail.com
{yaoli0508, xyuming, zhangdazhi}@hit.edu.cn

## Abstract

The transferability of adversarial perturbations provides an effective shortcut for black-box attacks. Targeted perturbations have greater practicality but are more difficult to transfer between models. In this paper, we experimentally and theoretically demonstrated that neural networks trained on the same dataset have more consistent performance in *High-Sample-Density-Regions* (HSDR) of each class instead of low sample density regions. Therefore, in the target setting, adding perturbations towards HSDR of the target class is more effective in improving transferability. However, density estimation is challenging in high-dimensional scenarios. Further theoretical and experimental verification demonstrates that easy samples with low loss are more likely to be located in HSDR. Perturbations towards such easy samples in the target class can avoid density estimation for HSDR location. Based on the above facts, we verified that adding perturbations to easy samples in the target class improves targeted adversarial transferability of existing attack methods. A generative targeted attack strategy named Easy Sample Matching Attack (**ESMA**) is proposed, which has a higher success rate for targeted attacks and outperforms the SOTA generative method. Moreover, ESMA requires only $5\%$ of the storage space and much less computation time comparing to the current SOTA, as ESMA attacks all classes with only one model instead of seperate models for each class. Our code is available at `https://github.com/gjq100/ESMA`

## 1 Introduction

Deep learning models exhibits substantial computational capacity in many downstream tasks, but are vulnerable to adversarial attacks [1, 2]. Such attacks tend to be transferable [3, 4] as well as real-world achievable [5], which makes it implementable in black-box scenarios. Targeted attacks are known to be more difficult to transfer [4, 6] compared with non-targeted attacks.

Intuitively, directions and transferability of adversarial attacks are closely related, but their relationship is rarely discussed. [7] found that samples in low density regions of ground truth distribution are more susceptible to adversarial attacks. They also verified that adversarial perturbations that aligned with the low density direction of ground truth distribution can lead to better transferability. For targeted attack scenarios, the direction that can bring more transferability has not been fully researched yet is not clear.

37th Conference on Neural Information Processing Systems (NeurIPS 2023).

**Findings and arguments.** In non-targeted scenarios, directing towards low-density regions of the ground-truth distribution can improve adversarial transferability [7]. However, this approach is not direct enough for targeted attacks. As shown in Figure 1, perturbations pointing at *High-Sample-Density-Regions* (*HSDR*) of the target domain are more effective than that pointing at low-density regions of the ground-truth distribution, it perturbates samples more directly to the target discrimination region. Moreover, we demonstrate and theoretically prove that neural networks tend to have more consistent outputs in the HSDR of each class, which means that perturbations pointing to the HSDR of the target class are also transferable between different models. However, density estimation for samples in high dimension is challenging.

Fortunately, by the definitions of hard samples and easy samples in [8], we found a fact that helps, in each category, easy samples with smaller losses are more likely to locate in HSDR. We provide theoretical and experimental assurance for this fact. Based on this fact, we can directly perturbate towards such easy samples of the target domain to improve transferability without density estimation.

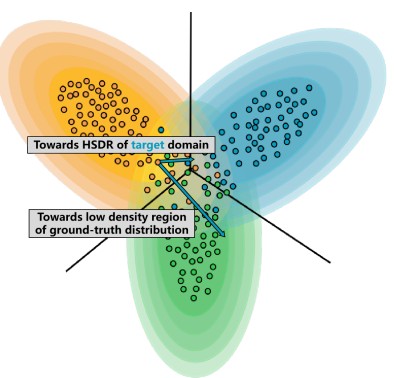

Figure 1: A schematic example of our motivation, plotting the probability density (darker the color represents larger the density) and samples for three populations (orange, cyan, and green). The black line indicates the Bayesian discriminant boundary.

**Related works.** Adversarial attacks can be divided into white-box attacks and black-box attacks. In the white-box setting, attackers have access to the model's structure and parameters, while in the black-box setting, they have no such information but only access to the input and output of the model. This is typically the scenario encountered in real-world situations. Black-box attacks include query-based attacks [9, 10, 11] and transfer-based attacks [12, 13, 14]. Conducting too many queries is impractical in real-world applications. In contrast, transfer-based attacks are more feasible as they do not require queries. Transfer-based attacks often require the use of a white-box surrogate model to produce adversarial perturbations. In terms of the way adversarial perturbations are generated, there are two approaches: iterative instance-specific methods and generative methods. Iterative instance-specific methods utilize model gradients to iteratively add perturbations to specified samples (e.g., FGSM [3], C&W [5], PGD [15]). To enhance the transferability of adversarial samples, subsequent work combines these methods with various techniques, such as introducing momentum [12, 13] and considering input transformations [13, 14, 16, 17] during iterations, training auxiliary classifiers [18, 19], or substituting different loss functions [20, 21, 22]. However, instance-specific methods are primarily designed for non-targeted scenarios, often lacking effectiveness in targeted settings. Although relatively good results have been achieved for targeted attacks [20, 21, 23], instance-specific methods still require iteratively creating perturbations for each specified sample, while generators trained for generating adversarial perturbations can be generalized on more samples after training [24, 25, 26, 27].

The leading perturbation generative method currently is TTP [24], which employs a target-specific GAN to align clean and augmented data from the source domain with the target domain data. However, TTP necessitate training a dedicated generator for each class. This significantly increases storage requirements and training time. In our work, we design a generator for simultaneous attacks on all target classes, which uses class embedding information for each target class. To construct these embeddings, we adopt techniques similar to SNE [28] to align the surrogate model's output logits for target classes with the generator's embeddings. This enables latent features learned by the surrogate model to guide the construction of well-structured class embeddings. Building on our findings, we train a multi-class perturbation generator to simultaneously perturb the source domain samples towards easy samples with low loss of each target class. Experiments on the ImageNet dataset shown that our method can obtain a better target transfer success rate than TTP, while requiring much less storage. Our method also has certain advantages in training time.

Therefore, our contribution can be summarized as follows:

- We found that deep learning models have more consistent outputs in HSDR of each class, as demonstrated theoretically and through experiments (Section 2.1). This implies that adding

adversarial perturbations pointing to the HSDR of the target class results in better targeted adversarial transferability.

- We experimentally and theoretically verified that easy samples with low loss in early-stopping models are likely to be located in HSDR, which allows us to directly add perturbations pointing to such samples of the target class to improve targeted adversarial transferability (Section 2.2). This avoids density estimation of high-dimensional samples, which is challenging and computationally expensive.

- We introduced the Easy Sample Matching Attack (ESMA) (Section 3). ESMA achieves a higher targeted transfer success rate compared to SOTA generative attacks TTP. Furthermore, it only needs one model to perform attacks for all target classes (Section 4), which requires much less storage space than TTP (only about 1/20 of the storage space in 10 targets case), and requires less training time.

Our findings and conclusions can not only provide guidance for target transfer attacks, but more importantly, they reveal the consistency of deep learning models in the HSDR, and the correlation between sample difficulty and local sample density. Not only that, the design of our multi-target perturbation generative model can provide a new reference for subsequent related research.

## 2 Main conclusions

In this section, we combine illustrative experiments and theoretical proofs to illustrate our two conclusions in turn, and we use the following notations and definitions. $\mathcal{Z} = \mathcal{X} \times \mathcal{Y} \in \mathbb{R}^d \times \mathbb{R}$ is the sample space. The i.i.d. dataset $S = \{x_i, y_i\}_{i=1}^n$ consists of $n$ sample pairs $(x_i, y_i), 1 \leq i \leq n$. Given $y$, the conditional distribution of $x$ is $\mathcal{D}_{x|y}$. Feature mapping $f : \mathcal{X} \to \mathbb{R}^K$, where $K$ is the number of class, the class of feature mapping $f \in \mathcal{F}$. Specifically, the parametrized class $\mathcal{F}_{\boldsymbol{w}} := \{f_{\boldsymbol{w}} : \mathcal{X} \to \mathbb{R}^K, \boldsymbol{w} \in \mathcal{W}\}$, where $\mathcal{W}$ is the parameter space. Define $\mathcal{F}_{\boldsymbol{w}}^{\mathcal{S}} := \{f_{\boldsymbol{w}}^{\mathcal{S}} = \mathcal{S} \circ f_{\boldsymbol{w}} : f_{\boldsymbol{w}} \in \mathcal{F}_{\boldsymbol{w}}, \mathcal{S} \circ f_{\boldsymbol{w}}(x) = \mathcal{S}(f_{\boldsymbol{w}}(x))\}$, $\mathcal{S}$ denotes Softmax-transformation. The Softmax-Cross-Entropy loss is denoted as $\ell_{sce}(\cdot, \cdot) : \mathbb{R}^K \times \mathbb{R} \to \mathbb{R}$. Let $S_j := \{(x, y) \in S : y = j\}, \mathcal{I}_j := \{i : (x_i, y_i) \in S^j\}, \mathcal{C}_j := \{x \in \mathcal{X} : (x, y) \in \mathcal{Z}, y = j\}$.

**Definition 1** $((j, x_0, r)$**-Local sample density)** *Given a class $j \in [K]$, $(x_0, y_0) \in S_j$, the $(j, x_0, r)$-Local sample density:*

$$\rho_{(j, x_0, r)} = \frac{\sum_{i \in \mathcal{I}_j} \mathbb{1}(x_i \in \mathcal{B}(x_0, r))}{\text{vol}\mathcal{B}(x_0, r)}$$

*where $\mathcal{B}(x_0, r) := \{x \in \mathbb{R}^d : \|x - x_0\| \leq r\}$, $\text{vol}\mathcal{B}(x_0, r)$ denotes the volume of $\mathcal{B}(x_0, r)$.*

Now we use the notation $\mathcal{I}_{(j, x_0, r)} = \{i : (x_i, y_i) \in S_j, x_i \in \mathcal{B}(x_0, r), (x_0, y_0) \in S_j)\}, \mathcal{C}_{(j, x_0, r)} = \{x : x \in \mathcal{C}_j, x_i \in \mathcal{B}(x_0, r), (x_0, y_0) \in S_j)\}$. For simplicity, we denote $\ell_{sce}(f_{\boldsymbol{w}}(x), y)$ as $\ell(\boldsymbol{w}, x)$ and define the local empirical risk $R_{(j, x_0, r)}(\boldsymbol{w}) = \frac{1}{|\mathcal{I}_{(j, x_0, r)}|} \sum_{i \in \mathcal{I}_{(j, x_0, r)}} \ell(\boldsymbol{w}, x_i)$.

### 2.1 The Output Consistency of Different Deep Learning Models in HSDR

We conduct an experiment to explain why samples in low-density regions of ground-truth distribution are vulnerable to attacks, and verify the consistency of the output of the deep learning model in HSDR. A dataset consist of 200 samples is constructed by sampling from two 2-d Gaussian distributions with equal probability, which is then used to train a neural network for classification. Subsequently, we plot the discriminant region of the Bayes' criterion (which has the minimum error rate) for the known ground truth prior distribution and compared it with the discriminant region of the trained classifier. In addition, we train two other neural networks with different structures and parameter quantities, and plot the output differences of the three neural networks (Figure 2). The discriminant region in Figure 2(a) can reach the Bayesian error rate. The intersection of the two population distributions in the middle of the probability density curve (as shown in Figure 2(b)) belongs to the low-density region of the ground-truth distribution and also to the misclassified region. In such a region, even with minimal expected error, the Bayesian discriminant criterion will classify samples with relatively small probability density of a single population into another class. These samples can be thought of as "outliers". The trained NN classifier discriminates samples into their original categories, causing a

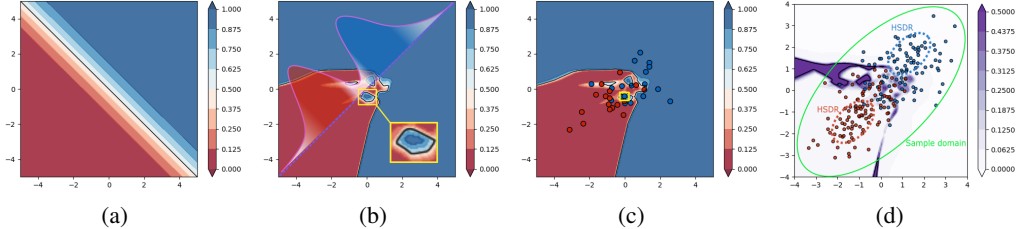

Figure 2: (a): Bayesian discriminant region, darker the color indicate higher the confidence probability. (b): Classifier discriminant region, the probability density curves of the two population distributions are plotted, the white part represents the low-density region of ground truth joint distribution, and we boxed out the small pits in the Bayesian misclassified region. (c): Classifier discriminant region with samples. We boxed an outlier. (d): Output differences between three different classifiers, darker purple indicates greater difference in output between different classifiers

difference in decision boundaries from the Bayesian prior classifier and creating small pits as shown in Figure 2(a). Most of the samples around the pit (Figure 2(c)) belong to another class due to its lower population density compared to the other class. Hence, perturbing the samples from this pit (i.e. outliers) towards the discriminant region of the other class becomes easier. If other trained neural networks can accurately classify such outliers, they will also generate similar pits. By adding perturbations pointing these pits to samples from another class, they will be perturbed into such pits, enabling the transfer of adversarial samples between different classifiers, which explained the findings of [7].

However, as we stated in Figure 1, perturbations towards the low-density regions of the ground-truth distribution are not direct enough for targeted attacks, while perturbations towards the HSDR of the target class are more direct. Combining this with Figure 2(d), we observe that different classifiers have more consistent outputs in the HSDR. With Theorem 1, we theoretically verified this, which also implies that perturbing samples to the HSDR of the target class leads to better targeted adversarial transferability.

**Theorem 1 (Local output consistency)** *For a target class $j \in [K]$, and two different parametrized class $\mathcal{F}_{\boldsymbol{w}_1} := \{f_{\boldsymbol{w}_1} : \boldsymbol{w}_1 \in \mathcal{W}_1\}$, $\mathcal{F}_{\boldsymbol{w}_2} := \{f_{\boldsymbol{w}_2} : \boldsymbol{w}_2 \in \mathcal{W}_2\}$, assume that $\frac{1}{|\mathcal{I}_{(j,x_0,r)}|} \left| \sum_{i \in \mathcal{I}_{(j,x_0,r)}} \left( f_{\boldsymbol{w}_1}^{\mathcal{S}_k}(x_i) - f_{\boldsymbol{w}_2}^{\mathcal{S}_k}(x_i) \right) \right| \leq \gamma$, then for any sample $(x_0, y_0) \in S_j$, in the neighborhood $\mathcal{B}(x_0, r)$, with probability at least $1 - \delta$, the following holds:*

$$\left\| \mathbb{E}_{x \sim \mathcal{D}_{x|y}} \left[ f_{\boldsymbol{w}_1}^{\mathcal{S}}(x) - f_{\boldsymbol{w}_2}^{\mathcal{S}}(x) \mid x \in \mathcal{C}_{(j,x_0,r)} \right] \right\|_{\infty}$$

$$\leq \mathcal{O} \left( \sqrt{\frac{K d^{d/2}}{\rho_{(j,x_0,r)} 2^d r^d} \log^2 \left( r^d \sqrt{\rho_{(j,x_0,r)}} \right)} + \sqrt{\frac{d^{d/2} \log(2K/\delta)}{\rho_{(j,x_0,r)} 2^{d+1} r^d}} \right) + \gamma.$$

**Remark.** Theorem 1 suggests that different models have a more consistent output near the samples in HSDR, specifically, the difference between the outputs has a bound of order $\tilde{O} \left( \sqrt{\frac{K d^{d/2}}{\rho_{(j,x_0,r)} 2^d r^d}} \right)$. When the number of classes and dimensions are fixed, a larger sample density results in more consistent performance, a weaker consistency within smaller neighborhoods. Note that this local consistency is weakened as the dimension increases, but the relative output consistency between HSDR and LSDR does not change.

However, a very practical problem is that sample points in high-dimensional space tend to be very discrete due to the *curse of dimentionality* [29], which makes it difficult to find a suitable local neighborhood size to calculate the local density of samples, moreover, in the case of large datasets, calculating local density can be computationally expensive. But the conclusion of the next section can help us find the sample located in the HSDR using early-stopping models without directly calculating the local density of the sample.

## 2.2 Correlation between Local Sample Density and Sample Difficulty

In this section we illustrate the correlation between sample density and sample difficulty. Hard samples have attracted much attention in various tasks because of their significance for training

convergence and model generalization [8, 30, 31, 32]. [8] proposed that for the convergent model, the difficulty of the sample can be measured by the loss gradient norm of the sample, easy samples has a relatively small loss gradient norm, while the loss gradient norm of difficult samples is relatively large, especially, those with too large gradient norm may be outliers.

We use the following theoretical analysis to illustrate that for a trained classifier, the samples in the HSDR tend to have a smaller local empirical risk. At the same time, smaller loss gradient norms ensure that losses are more consistent in small neighborhoods, which implies those samples with both smaller loss and loss gradient norms guarantee less local empirical risk in the neighborhood where the sample is located. To illustrate the following conclusions, we make several mild assumptions:

**Assumption 1 (Smoothness assumption)** $\ell(\boldsymbol{w}, x)$ *satisfies the following conditions of Lipschitz continuous gradient:*

$$\|\nabla_{\boldsymbol{w}}\ell(\boldsymbol{w}_1, x) - \nabla_{\boldsymbol{w}}\ell(\boldsymbol{w}_2, x)\| \le L_1 \|\boldsymbol{w}_1 - \boldsymbol{w}_2\|, \forall \boldsymbol{w}_1, \boldsymbol{w}_2 \in \mathcal{W},$$
$$\|\nabla_x \ell(\boldsymbol{w}, x_1) - \nabla_x \ell(\boldsymbol{w}, x_2)\| \le L_2 \|x_1 - x_2\|, \forall x_1, x_2 \in \mathcal{X}.$$

**Assumption 2** $\|\nabla_{\boldsymbol{w}}\ell(\boldsymbol{w}, x)\| \le G$ *for all* $\boldsymbol{w} \in \mathcal{W}$.

**Assumption 3 (Polyak-Łojasiewicz Condition)** $R_{(j,x_0,r)}(\boldsymbol{w})$ *satisfies the PL-condition:*

$$\frac{1}{2}\left\|\nabla_{\boldsymbol{w}}R_{(j,x_0,r)}(\boldsymbol{w})\right\|^2 \ge \mu\left(R_{(j,x_0,r)}(\boldsymbol{w}) - R^*_{(j,x_0,r)}\right),$$

*where* $R_{(j,x_0,r)}(\boldsymbol{w}) = \frac{1}{|\mathcal{I}_{(j,x_0,r)}|}\sum_{i \in \mathcal{I}_{(j,x_0,r)}} \ell(\boldsymbol{w}, x_i)$ *is the local empirical risk,* $R^*_{(j,x_0,r)} = \inf_{\boldsymbol{w} \in \mathcal{W}} R_{(j,x_0,r)}(\boldsymbol{w})$.

**Assumption 4** *For* $\forall i \in \mathcal{I}_{(j,x_0,r)}$, *the following holds:*

$$\left\langle \nabla_{\boldsymbol{w}}R_{(j,x_0,r)}(\boldsymbol{w}), \nabla_{\boldsymbol{w}}\ell(\boldsymbol{w}, x_i)\right\rangle \ge \beta \left\|\nabla_{\boldsymbol{w}}R_{(j,x_0,r)}(\boldsymbol{w})\right\|^2$$

Assumption 1 and 2 were made in [33], [34] and [35], where Assumption 2 can actually be derived from Assumption 1 when the input space is bounded, which is usually satisfied. Even a non-convex function can still satisfy Assumption 3 [34], the inequality in Assumption 3 implies that all stationary points are global minimum [36], which is proved in recently works [37, 38] for over-parameterized DNNs. Assumption 4 is reasonable when local neighborhood radius $r$ is small.

---

**Algorithm 1** Mini-batch SGD

---

**Require:** Initialized weights $\boldsymbol{w}^1$, total steps $T$, sample set $S$, batch size $M$ and step size $\eta_t$.
1: **for** $t = 1 \leftarrow T$ **do**
2:    Randomly sample $M$ different samples $B^t$ from $S$, where batch size $|B_t| = M$, corresponding indicator set denote as $\mathcal{I}_{B^t}$.
3:    $\boldsymbol{w}^{t+1} = \boldsymbol{w}^t - \frac{1}{M}\sum_{i \in \mathcal{I}_{B^t}} \nabla_{\boldsymbol{w}}\ell(\boldsymbol{w}^t, x_i)$
4: **end for**
**Return:** $\boldsymbol{w}^{T+1}$

---

Consider the local empirical risk under Algorithm 1, we use Theorem 2 to illustrate the convergence rate relies on local density.

**Theorem 2 (Optimization relies on local density)** *Given a learnable parametrized class* $\mathcal{F}_{\boldsymbol{w}} := \{f_{\boldsymbol{w}} : \mathcal{X} \to \mathbb{R}^K, \boldsymbol{w} \in \mathcal{W}\}$, *let* $\boldsymbol{w}^t$ *updated by Algorithm 1, under Assumption 1, 2, 3 and 4, set* $\eta_t = \frac{1}{\beta\mu t}$ *and* $T \le \frac{L_1}{2\beta\mu}$, *then with probability at least* $1 - \delta$, *holds*

$$R_{(j,x_0,r)}(\boldsymbol{w}^{t+1}) - R^*_{(j,x_0,r)} \le \left(1 - \frac{2}{t}\tau\left(\rho_{(j,x_0,r)}\right)\right)\left(R_{(j,x_0,r)}(\boldsymbol{w}^t) - R^*_{(j,x_0,r)}\right) + o\left(\frac{1}{t^2}\right),$$

*where* $\tau\left(\rho_{(j,x_0,r)}\right) = \max\left\{\left(\frac{\rho_{(j,x_0,r)}\pi^{d/2}r^d}{\Gamma(\frac{d}{2}+1)M} - \sqrt{\frac{\ln(T/\delta)}{2M}}\right), 0\right\}$, *which is a non-descending function of* $\rho_{(j,x_0,r)}$.

According to theorem 2, the local empirical risk $R_{(j,x_0,r)}(\boldsymbol{w})$ will reach a more faster convergence rate in HSDR and, relatively, a relatively slower convergence rate in LSDR, thus for early-stopping models, they has a lower local empirical risk in HSDR. Combined with the following Proposition 1, we show that a smaller gradient norm guarantees a smaller local empirical risk. For overfitting models, the local empirical risk of each neighborhood where samples are located may be very small, but for early-stopping models, this relativity of local empirical risk with respect to local sample density can be maintained.

**Proposition 1** *Under Assumption 1, given any $(x_i, y_i) \in S$, for any $x \in \mathcal{X}$ that satisfies $\|x_i - x\| \leq r$, the following holds:*

$$\frac{|\ell(\boldsymbol{w}, x_i) - \ell(\boldsymbol{w}, x)|}{r} - \frac{3L_2 r}{2} \leq \|\nabla_x \ell(\boldsymbol{w}, x_i)\| \leq \frac{\ell(\boldsymbol{w}, x_i)}{r} + \frac{3L_2 r}{2}.$$

**Remark.** Proposition 1 suggests that minimizing the loss leads to a smaller norm of the loss gradient. However, this constraint becomes less tight as the neighborhood radius $r$ decreases. Compared with the loss itself, the loss gradient norm further guarantees the proximity of local loss, especially when $r$ is small. Therefore, samples with smaller loss and smaller loss gradient norm are more likely to be in a neighborhood with smaller local empirical risk.

Using the above conclusion, for an early-stopping model, samples with smaller losses and loss gradient norms tend to be located in HSDR. Still in the example in the previous section, we train three models with early-stopping, and the learning rate adjusted with the number of steps, and then plotted Figure 3. The model has more consistent outputs in HSDR. When the loss and gradient norms of a sample are small, the region where the sample is located has smaller local empirical risk. Samples in HSDR have smaller local empirical risk, which is consistent with our theoretical analysis. Additionally, the rightmost graph of Figure 3 shows that samples with smaller loss and gradient norms often locate in HSDR, validating our conclusions.

Therefore, we can determine whether a sample is more likely to located in HSDR or LSDR by evaluating whether it has smaller loss and gradient norms simultaneously. This eliminates the need to calculate local sample densities to find samples in HSDR.

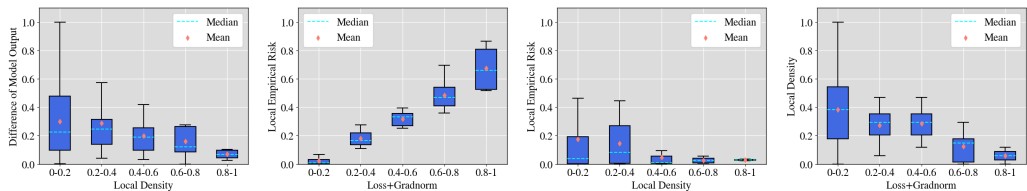

Figure 3: The first figure depicts the difference in output of three models under different local sample densities $\rho_{(y_i, x_i, r)}$ divided into different bins. The second figure shows the local empirical risk $R_{(y_i, x_i, r)}$ of samples under different sum of loss and gradient norms (**Loss+Gradnorm**). For Loss+Gradnorm, we first normalize both variables separately and then add them up to eliminate magnitude differences. The third figure represents the local empirical risk of local sample densities in different values. The fourth figure displays the local density under different Loss+Gradnorms. The neighborhood radius $r$ is taken as $0.4$.

We conduct transfer attack experiments on three baselines on the CIFAR10 dataset, three different models are chosen as victims. Combining our perspectives above, we use algorithm 2 to select the anchor of each target class for guiding the addition of adversarial perturbations. We implement our strategy by simply using squared loss to match anchor points, i.e. minimizing $\left\| f(x_i^{adv}) - a_{\text{target}_i} \right\|^2$ (choosing $q = 10$ and $\epsilon = 16$ pixels). For comparison, we also use cross-entropy (CE) loss to calculate adversarial examples in the vanilla way, as well as using squared error without a screening mechanism (i.e., randomly selecting the target anchor in the target class) to exclude the influence of different losses. The results are shown in Table 1, our strategy indeed helps to enhance the transferability of target attacks, which further confirms our viewpoints.

**Algorithm 2** Target Anchor Screening

**Require:** Early-stopping classifier $f_{\boldsymbol{w}}$, screening parameter $q$ and sample set $S$.

1: **for** $i = 1 \leftarrow n$ **do**
2: $\quad \text{loss}_i = \ell_{sce}(f(x_i), y_i), \text{gradnorm}_i = \|\nabla_x \ell_{sce}(f(x_i), y_i)\|.$
3: **end for**
4: For each class $k \in [K]$, select the $q$-th smallest loss and the gradient norm among the samples in that class as thresholds $\text{thr}_k^{\text{loss}}$ and $\text{thr}_k^{\text{gradnorm}}$
5: **for** $k = 1 \leftarrow K$ **do**
6: $\quad A_k := \left\{ i \in \mathcal{I}_k : \text{loss}_i < \text{thr}_k^{\text{loss}}, \text{gradnorm}_i < \text{thr}_k^{\text{gradnorm}} \right\},$
$\quad a_k = \frac{1}{|A_k|} \sum_{j \in A_k} f_{\boldsymbol{w}}(x_j):$
7: **end for**

Table 1: Targeted transfer success rates. "left/middle/right" represent attacks using regular CE loss, square loss with randomly selected target anchors, and square loss with target anchor screening, respectively. In parentheses, the clean accuracy is indicated.

| Attack | Src:Res34(95.44%) | | Src:VGG16(94.27%) | | Src:Dense121(95.47%) | |
|---|---|---|---|---|---|---|
| | →VGG16 | →Dense121 | →Res34 | →Dense121 | →Res34 | →VGG16 |
| MIM | 14.32%/12.94%/**14.44**% | 19.81%/19.00%/**20.09**% | 14.80%/13.78%/**15.13**% | 13.41%/12.11%/**13.83**% | 17.13%/14.17%/**17.17**% | 11.23%/10.27%/**11.25**% |
| TIM | 13.53%/12.50%/**13.88**% | 15.23%/14.56%/**15.71**% | 12.49%/11.78% /**12.88**% | 11.14%/ 10.17% /**11.40**% | 15.74%/14.61%/**15.89**% | 11.93%/10.94%/**12.09**% |
| DIM | 15.46%/14.39%/**15.97**% | 18.10%/ 17.61%/**18.96**% | 14.56%/13.28%/**15.12**% | 12.97%/12.11%/**13.47**% | 18.11%/16.83%/**18.33**% | 13.18%/11.72%/**13.26**% |

## 3 Training Strategy of ESMA

We present our training strategy in this section, as shown in Figure 4. Our training strategy is carried out in two steps, the first step we pre-train the generator's embedding representations, and the second step is to find easy samples of each target class based on our previous conclusions, and then use these samples to guide the generator to generate perturbations from the source domain to the target domain.

**Pre-trained Embeddings Guided by Latent Features** To obtain better embeddings that more effectively represents inter-class information, since the latent space of deep learning models often extracts enough class information [39], we treat the output features of the local model as a set of a priori embedding. In order to make the generator embedding have a similar structure to such a priori embedding, we refer to the idea of manifold learning, using a strategy similar to the SNE algorithm [28]. Let $\mu_j = \frac{1}{|\mathcal{I}_j|} \sum_{i \in \mathcal{I}_j} l_i$, where $l_i = f(x_i)$, and then we design the following manifold

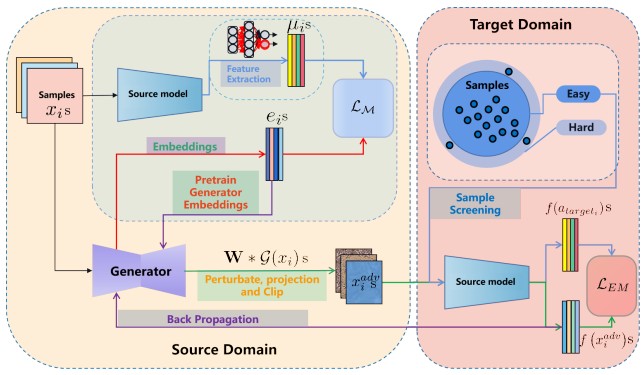

Figure 4: Training strategy of ESMA.

matching loss. Generator embeddings of various class were pulled to the manifold that output features in to obtain embeddings with a better structure.

Next, we denote the generator embedding of class $j$ as $e_j$. The four matrices $M^{S_{euc}}, M^{E_{euc}}, M^{S_{cos}}, M^{E_{cos}}$ satisfy $M^{S_{euc}}_{i,j} = \|\mu_i - \mu_j\|$, $M^{E_{euc}}_{i,j} = \|e_i - e_j\|$, $M^{S_{cos}}_{i,j} = \frac{\mu_i \cdot \mu_j}{\|\mu_i\|\|\mu_j\|}$, $M^{E_{cos}}_{i,j} = \frac{e_i \cdot e_j}{\|e_i\|\|e_j\|}$, respectively. Let

$$\overline{M}^{S_{\text{euc}}}_{i,j} = \frac{\exp\left(M^{S_{\text{euc}}}_{i,j}\right)}{\sum_{k=1}^{K} \exp\left(M^{S_{\text{euc}}}_{i,k}\right)}, \overline{M}^{E_{\text{euc}}}_{i,j} = \frac{\exp\left(M^{E_{\text{euc}}}_{i,j}\right)}{\sum_{k=1}^{K} \exp\left(M^{E_{\text{euc}}}_{i,k}\right)},$$
$$\overline{M}^{S_{\text{cos}}}_{i,j} = \frac{\exp\left(M^{S_{\text{cos}}}_{i,j}\right)}{\sum_{k=1}^{K} \exp\left(M^{S_{\text{cos}}}_{i,k}\right)}, \overline{M}^{E_{\text{cos}}}_{i,j} = \frac{\exp\left(M^{E_{\text{cos}}}_{i,j}\right)}{\sum_{k=1}^{K} \exp\left(M^{E_{\text{cos}}}_{i,k}\right)},$$

then our manifold matching loss is as follows:

$$\mathcal{L}_{\mathcal{M}} = \sum_{i,j} \overline{M}_{i,j}^{S_{\mathrm{euc}}} \log \frac{\overline{M}_{i,j}^{S_{\mathrm{euc}}}}{\overline{M}_{i,j}^{E_{\mathrm{euc}}}} + \sum_{i,j} \overline{M}_{i,j}^{E_{\mathrm{euc}}} \log \frac{\overline{M}_{i,j}^{E_{\mathrm{euc}}}}{\overline{M}_{i,j}^{S_{\mathrm{euc}}}}$$
$$+ \lambda_1 \left[ \sum_{i,j} \overline{M}_{i,j}^{S_{\mathrm{cos}}} \log \frac{\overline{M}_{i,j}^{S_{\mathrm{cos}}}}{\overline{M}_{i,j}^{E_{\mathrm{cos}}}} + \sum_{i,j} \overline{M}_{i,j}^{E_{\mathrm{cos}}} \log \frac{\overline{M}_{i,j}}{\overline{M}_{i,j}} \right] + \lambda_2 \sum_{i=1}^{K} \left\| e^i \right\|.$$

The last regular term is to prevent losses from collapsing. $\lambda_1$ and $\lambda_2$ are hyperparameters. The pre-trained embedding with our strategy has a larger Euclidean distance and a smaller cosine similarity, which greatly alleviates the previous clustering phenomenon. More detailed analysis and discussion are provided in Appendix B.

**Training of Multi-target Adversarial Perturbation Generators**   After pre-training embedding, we freeze the parameters of the embedding layer, combined with our previous conclusions, we select several easy samples in each class to form target anchor sets $A_i$, and match them with the output of our generator in feature space, the generator we use is a Unet with Resblocks. We propose the following objective easy sample feature matching loss to train the multi-class adversarial generator.

$$\mathcal{L}_{EM} = \sum_{j=1}^{K} \frac{1}{\sum_{i \in [n]} \mathbb{1} \left( i \notin \mathcal{I}_j \right)} \sum_{i \notin \mathcal{I}_j} d \left( f \left( a_j \right), f \left( \mathrm{clip}_\epsilon \left( \mathbf{W} * \mathcal{G}_\theta \left( x_i \right) \right) \right) \right),$$

where $\mathrm{clip}_\epsilon(x) = \mathrm{clip} \left( \min \left( x + \epsilon, \max \left( x, x - \epsilon \right) \right) \right)$, $\mathbf{W}$ is a differentiable Gaussian kernel with size $3 * 3$, such smoothing operations have been demonstrated to further improve transferability[24]. The measure of the distance between the two features $d(\cdot, \cdot)$ is Smooth L1 loss, which has a unique optimal solution that is not sensitive to exceptional values [40]. Then, our final training strategy can be represented by algorithm 3.

---

**Algorithm 3** Training Strategy of ESMA

---

**Require:** Generator $\mathcal{G}_\theta$ with pre-trained embeddings, anchors $a_k, k \in [K]$ and Total epochs $N$.
1: **for** $t = 1 \leftarrow N$ **do**
2:   **for** $i = 1 \leftarrow n$ **do**
3:     $\mathrm{target}_i \sim \mathrm{Uniform}(\{1, \ldots, K\})$ .
4:     **if** $y_i \neq \mathrm{target}_i$ **then**
5:       Random choice an anchor $a_j$ from $A_j$,
6:       Gradient descent step on $\mathcal{L}_{EM}$.
7:     **end if**
8:   **end for**
9: **end for**

---

## 4 Experiments

In this section, we verify the effectiveness of our method through experiments on ILSVRC2012 dataset [41]. To evaluate the effectiveness of different components in our strategy, we conduct ablation experiments in Section 4.3. Other ablation experiments can be found in Appendix C.

### 4.1 Experiment Setup

**Dataset**   The dataset we used comes from the ILSVRC2012 dataset [41], in which we selected ten classes as the training set, which refer to the ten classes used for TTP training in [24], they are 24, 99, 198, 245, 344, 471, 661, 701, 802, 919. We train generators using images of these ten classes in the training set (1300 images per class), and use the images in the validation set (50 images per class) as the validation dataset for the targeted attack.

**Models**   We use the four networks used in [24] as source models —ResNet-50 [42] (Res50), VGG-19-bn [43] (VGG19bn), DenseNet-121 [44] (Dense121), ResNet-152 [42] (Res152). Except the above four models, we also select three models from the Inception series: Inception-v3 [45] (Inc-v3), Inception-v4 [46] (Inc-v4), Inception-ResNet-v2 [46] (IncRes-v2) and a transformer vision model, VIT [47]. As the analysis and fundamental assumptions of this paper are based on the condition of training data from the same distribution, in order to explore the transferability between models trained on training data with different distribution, we conduct additional transfer attack on two

adversarially trained models, Inc-v3-adv [48] and IncRes-v2-ens [49]. Results are shown in E.2. In addition, we also test the adversarial transferability of ESMA on the scenario where the source model is an ensemble of different models, corresponded results are reported in E.1.

**Baselines** We select many iterative instance-specific attack benchmarks, MIM [12], SI-NIM [13], TIM [14], DIM [16], and advanced iterative instance-specific attacks that are competitive in target setting, Po-TI-Trip [20], Logit [21], Rap-LS [23], FGS$^2$M [50], DMTI-Logit-SU [51], S$^2$I-SI-TI-DIM [52]. Generative adversarial attacks HGN [53] and TTP [24] also included in our comparision, where TTP is the current SOTA generative method. As a generative attack, TTP requires training a class-dependent generator specifically for each class.

**Parameter Setting** For the parameter settings of different attack methods, we refer to the default settings in [20], total iteration number $T = 20$, step size $\alpha = \epsilon/T$, where the $\ell_\infty$ perturbation restriction $\epsilon$ is set to 16. Momentum factors $\mu$ is set to 1, for the stochastic input diversity in DIM, we set the probability of applying input diversity as 0.7. For TIM, the kernel-length is set to 7, which is more suitable for targeted attacks. For Po-TI-Trip, the weight of triplet loss $\lambda$ is set to 0.01, while the margin $\gamma$ is set to 0.007. Referring to [21], the number of iteration steps of Logit is set to 300, and for RAP-LS, we choose 400 iteration steps, $K_{LS}$ is set to 100, and $\epsilon_n$ is set to $12/255$ [23]. Then for TTP, since we used a relatively small training set, we added 10 epochs to the original paper [24] settings to ensure the performance of the model, and the learning rate of Adam optimizer is $1e-4$ ($\beta_1 = .5$, $\beta_2 = .999$). Finally, for our model, we used the AdamW optimizer to train 300 epochs with a learning rate of $1e-4$ (350 for cases where the source model is VGG19bn or Dense121), the value of $q$ used for sample screening is set to 2.

**Evaluation Setting** We train the generator on the training set of the selected ten classes, verify the targeted transferality on the validation set, for each target class, we use the 450 images of the remaining classes as the source data and perturb them to the target class. For the iterative instance-specific attacks, we directly attack these instances and test their targeted transfer success rate. All methods (including training) were implemented on a single NVIDIA RTX A5000 GPU.

## 4.2 Results

The results of our experiments, reported in Table 3, ESMA outperforms current SOTA generative attack TTP in targeted transfer success rates. Our approach demonstrates significant advantages in terms of efficiency and effectiveness, with TTP having a parameter count of 7.84M per model compared to ESMA's 4.40M. Additionally, ESMA achieves an average training time that is 78.4% of TTP. The experimental results also strongly supports our viewpoint.

## 4.3 Ablation Studies

To validate the effectiveness of the two components in our strategy design, we compared the cases with (w/) and without (w/o) pre-trained embeddings and target anchor screening. The results are shown in Figure 5. In the case without pre-trained embeddings, the performance is significantly weaker than the case with pre-trained embeddings across different training durations. Moreover, the case with target anchor screening shows a noticeable improvement compared to randomly selecting target anchors.

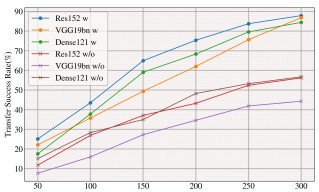

Figure 5: Comparison of targeted attack transfer success rates with (w) pre-trained embeddings and without (w/o) pre-trained embeddings at different training epochs. Src:Res50.

To verify that ESMA can indeed increase the sample density of the target class for the original samples, we use the samples used for adversarial testing in table 3. First, we calculate the sample density of the target class for the clean samples $\rho_{(\text{target},x_i,r)}$ (here $r$ is set to 600). Then, we apply ESMA to generate adversarial samples $x_i^{adv}$, and calculate the sample density of the target class for these adversarial samples $\rho_{(\text{target},x_i^{adv},r)}$. In both cases, the density calculation results were averaged across all target classes for each sample. We normalize the results by dividing them by the maximum value among all results in both cases. Additionally, we further bin and count the number of test samples in different intervals under different perturbation constraints. The results are shown in Table 2. The

target class sample density of the samples perturbated using ESMA has increased in bins with larger values, which confirms that ESMA can indeed enhance the target class sample density.

Table 2: Comparison of sample density (binned).

| | |Counts| | 0-0.1 | 0.1-0.2 | 0.2-0.3 | 0.3-0.4 | 0.4-0.5 | 0.5-0.6 | 0.6-0.7 | 0.7-0.8 | 0.8-0.9 | 0.9-1.0 |
|---|---|---|---|---|---|---|---|---|---|---|---|---|
| Case for $\epsilon = 8/255$: | Clean | 32 | 35 | 31 | 45 | 39 | 51 | 57 | 51 | 61 | 48 |
| | ESMA | 27 | 30 | 32 | 37 | 43 | 49 | 62 | 51 | 70 | 49 |
| Case for $\epsilon = 16/255$: | Clean | 32 | 35 | 31 | 45 | 39 | 51 | 57 | 51 | 61 | 48 |
| | ESMA | 25 | 27 | 36 | 30 | 49 | 50 | 59 | 54 | 71 | 49 |

Table 3: Targeted transfer success rates. "Src" indicates the source model. For TTP and ESMA, the numbers in parentheses indicate training duration in seconds.

| Src | Attack | →VGG19bn | →Dense121 | →Res152 | →Inc-v3 | →Inc-v4 | →IncRes-v2 | →ViT | AVG |
|---|---|---|---|---|---|---|---|---|---|
| **Res50** | MIM | 1.33% | 3.44% | 3.67% | 0.51% | 0.31% | 0.22% | 0.16% | 1.38% |
| | TIM | 13.07% | 26.16% | 23.82% | 5.51% | 3.84% | 2.96% | 0.60% | 10.85% |
| | DIM | 13.36% | 24.96% | 23.44% | 5.11% | 3.80% | 2.29% | 0.47% | 10.49% |
| | SI-NIM | 1.07% | 1.91% | 2.00% | 0.40% | 0.44% | 0.29% | 0.07% | 0.88% |
| | Po-TI-Trip | 17.93% | 33.62% | 32.20% | 7.73% | 5.69% | 3.38% | 1.07% | 14.52% |
| | SI-FGS$^2$M | 20.75% | 32.68% | 30.55% | 6.30% | 4.42% | 2.92% | 0.91% | 14.08% |
| | S$^2$I-SI-TI-DIM | 26.49% | 36.36% | 36.49% | 14.87% | 12.67% | 10.02% | 1.18% | 19.72% |
| | Logit | 51.60% | 77.22% | 76.98% | 17.62% | 12.67% | 8.40% | 3.09% | 35.37% |
| | DTMI-Logit-SU | 53.36% | 78.91% | 79.00% | 18.56% | 13.36% | 8.71% | 3.32% | 36.46% |
| | RAP-LS | 60.04% | 83.40% | 80.38% | 22.16% | 17.24% | 10.82% | 3.80% | 39.69% |
| | HGN | 63.92% | 70.64% | 68.43% | 22.41% | 13.76% | 7.31% | 12.12% | 36.94% |
| | TTP(95339.7) | 67.11% | 73.67% | 72.64% | 33.49% | 25.27% | 11.27% | 18.07% | 43.07% |
| | ESMA(82060.1) | **81.29%** | **83.89%** | **81.78%** | **37.53%** | **39.29%** | **17.29%** | **21.25%** | **51.76%** |

| Src | Attack | →VGG19bn | →Res50 | →Res152 | →Inc-v3 | →Inc-v4 | →IncRes-v2 | →ViT | AVG |
|---|---|---|---|---|---|---|---|---|---|
| **Dense121** | MIM | 1.98% | 2.33% | 1.51% | 0.56% | 0.40% | 0.27% | 0.27% | 1.05% |
| | TIM | 8.98% | 12.82% | 8.87% | 3.44% | 3.31% | 1.91% | 0.71% | 5.72% |
| | DIM | 10.20% | 13.80% | 9.18% | 3.98% | 3.27% | 2.29% | 0.60% | 6.19% |
| | SI-NIM | 8.29% | 11.42% | 6.96% | 1.78% | 1.71% | 1.13% | 0.29% | 4.51% |
| | Po-TI-Trip | 11.29% | 16.80% | 11.78% | 5.49% | 4.84% | 2.96% | 0.96% | 7.73% |
| | SI-FGS$^2$M | 12.55% | 16.67% | 11.70% | 5.16% | 4.89% | 3.58% | 0.85% | 7.91% |
| | S$^2$I-SI-TI-DIM | 25.60% | 30.33% | 23.98% | 13.11% | 12.04% | 7.93% | 1.38% | 16.33% |
| | Logit | 31.98% | 43.49% | 31.71% | 12.91% | 11.51% | 7.16% | 3.27% | 20.29% |
| | DTMI-Logit-SU | 33.38% | 44.93% | 34.16% | 13.44% | 12.36% | 7.75% | 3.62% | 21.38% |
| | RAP-LS | 38.56% | 50.67% | 38.24% | 15.78% | 13.96% | 9.64% | 3.82% | 24.38% |
| | HGN | 47.81% | 56.96% | 44.21% | 22.75% | 19.56% | 9.42% | 10.62% | 30.19% |
| | TTP(109644.6) | 52.00% | 58.02% | 49.24% | 29.69% | 23.00% | 13.24% | 17.71% | 34.70% |
| | ESMA(96335.6) | **56.83%** | **63.71%** | **54.70%** | **33.23%** | **29.28%** | **15.97%** | **18.50%** | **38.89%** |

| Src | Attack | →Res50 | →Dense121 | →Res152 | →Inc-v3 | →Inc-v4 | →IncRes-v2 | →ViT | AVG |
|---|---|---|---|---|---|---|---|---|---|
| **VGG19bn** | MIM | 0.56% | 0.58% | 0.29% | 0.27% | 0.20% | 0.04% | 0.07% | 0.29% |
| | TIM | 3.80% | 4.58% | 1.84% | 1.47% | 1.09% | 0.56% | 0.16% | 1.93% |
| | DIM | 2.98% | 4.33% | 1.76% | 1.09% | 1.16% | 0.47% | 0.16% | 1.71% |
| | SI-NIM | 0.42% | 0.42% | 0.29% | 0.11% | 0.20% | 0.18% | 0.07% | 0.24% |
| | Po-TI-Trip | 4.56% | 6.27% | 2.42% | 1.69% | 1.47% | 0.87% | 0.27% | 2.51% |
| | SI-FGS$^2$M | 4.12% | 5.71% | 1.97% | 1.56% | 1.25% | 0.68% | 0.25% | 2.22% |
| | S$^2$I-SI-TI-DIM | 12.09% | 14.84% | 6.49% | 4.87% | 5.80% | 2.93% | 0.31% | 6.76% |
| | Logit | 22.16% | 30.47% | 11.51% | 5.51% | 6.71% | 1.91% | 0.98% | 11.32% |
| | DTMI-Logit-SU | 23.47% | 31.47% | 12.60% | 6.13% | 7.31% | 2.07% | 1.11% | 12.02% |
| | RAP-LS | 24.31% | 33.33% | 13.56% | 6.58% | 8.51% | 2.78% | 1.11% | 12.88% |
| | HGN | 34.18% | 31.00% | 19.36% | 11.16% | 8.87% | 1.71% | 2.26% | 15.51% |
| | TTP(131519.2) | 36.76% | 34.44% | 22.11% | 11.82% | 10.40% | 2.20% | 2.58% | 17.19% |
| | ESMA(110769.2) | **39.61%** | **47.25%** | **22.85%** | **12.98%** | **11.73%** | **4.09%** | **3.35%** | **20.27%** |

| Src | Attack | →VGG19bn | →Res50 | →Dense121 | →Inc-v3 | →Inc-v4 | →IncRes-v2 | →ViT | AVG |
|---|---|---|---|---|---|---|---|---|---|
| **Res152** | MIM | 1.11% | 5.69% | 3.29% | 0.69% | 0.31% | 0.18% | 0.18% | 1.64% |
| | TIM | 9.76% | 26.64% | 21.56% | 5.69% | 4.22% | 3.04% | 1.07% | 10.28% |
| | DIM | 9.89% | 26.27% | 20.98% | 5.69% | 4.11% | 2.44% | 0.71% | 10.01% |
| | SI-NIM | 0.82% | 1.76% | 1.16% | 0.36% | 0.20% | 0.24% | 0.11% | 0.66% |
| | Po-TI-Trip | 14.53% | 35.36% | 29.87% | 8.93% | 6.29% | 4.71% | 1.36% | 14.44% |
| | SI-FGS$^2$M | 13.98% | 39.22% | 24.80% | 7.98% | 6.11% | 4.31% | 0.95% | 13.91% |
| | S$^2$I-SI-TI-DIM | 19.29% | 36.64% | 30.78% | 14.36% | 11.42% | 10.84% | 1.29% | 17.80% |
| | Logit | 35.44% | 75.22% | 61.31% | 15.33% | 10.69% | 8.11% | 2.76% | 29.84% |
| | DTMI-Logit-SU | 37.76% | 77.47% | 62.91% | 16.02% | 11.51% | 8.58% | 3.02% | 31.04% |
| | RAP-LS | 42.36% | 83.20% | 69.67% | 18.82% | 14.33% | 10.36% | 3.20% | 34.56% |
| | HGN | 61.40% | 73.31% | 67.89% | 23.47% | 26.87% | 11.31% | 13.96% | 41.17% |
| | TTP(132223.9) | 65.31% | 79.73% | 74.93% | 36.73% | 30.11% | 13.44% | 15.62% | 45.12% |
| | ESMA(93977.6) | **78.67%** | **88.18%** | **79.93%** | **41.68%** | **34.38%** | **14.58%** | **18.72%** | **50.88%** |

## 5 Conclusion

In this work, we provided a novel perspective on the transferability of targeted adversarial attacks. The study theoretically and experimentally demonstrated that adding perturbations towards HSDR of the target class can further enhance the transferability of targeted attacks. We also proposed a method for identifying samples within HSDR, which avoided the impracticality of density estimation in high-dimensional data. Building upon these insights, we introduced an improved generative multi-target attack strategy ESMA, which surpassed previous generative targeted attacks in both effectiveness and efficiency. We believe that our insights not only provide guidance for targeted attacks but also offer insights for dataset selection. Further discussion on these points would be included in our future work.

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

# *Supplementary Material*:
## Perturbation Towards Easy Samples Improves Targeted Adversarial Transferability

## A Proofs

### A.1 Proof of Theorem 1

First, we need the definition below:

**Definition 2** $((j, k, x_0, r)$-**Local Output Rademacher Complexity)** *Let* $f_{\boldsymbol{w}}^{\mathcal{S}} = (f_{\boldsymbol{w}}^{\mathcal{S}_1}, \ldots, f_{\boldsymbol{w}}^{\mathcal{S}_K}) \in [0,1]^K$, $\mathcal{F}_{\boldsymbol{w}}^{\mathcal{S}_k} := \{f_{\boldsymbol{w}}^{\mathcal{S}_k} : \boldsymbol{w} \in \mathcal{W}\}$, *given* $\mathcal{B}(x_0, r) = \{x : \|x - x_0\| \leq r\}$, $(x_0, y_0) \in S_j$, *then the empirical* $(j, k, x_0, r)$-*local output Rademacher Complexity is defined by:*

$$\mathcal{R}_n^{(j,x_0,r)}(\mathcal{F}_{\boldsymbol{w}}^{\mathcal{S}_k}) := \mathbb{E}_{\boldsymbol{\sigma}} \left[ \sup_{f_{\boldsymbol{w}}^{\mathcal{S}_k} \in \mathcal{F}_{\boldsymbol{w}}^{\mathcal{S}_k}} \left| \sum_{i \in \mathcal{I}_{(j,x_0,r)}} \sigma_i f_{\boldsymbol{w}}^{\mathcal{S}_k}(x_i) \right| \right],$$

*where* $\sigma_i, i \in \mathcal{I}_{(j,x_0,r)}$ *are i.i.d random variables satisfies* $\mathbb{P}(\sigma_i = 1) = \mathbb{P}(\sigma_i = -1) = \frac{1}{2}$. *Correspondingly, the expected* $(j, k, x_0, r)$-*local output Rademacher complexity is defined as* $\mathcal{R}^{(j,x_0,r)}(\mathcal{F}_{\boldsymbol{w}}^{\mathcal{S}_k}) = \mathbb{E}_{S_{(j,x_0,r)}} \left[ \mathcal{R}_n^{(j,x_0,r)}(\mathcal{F}_{\boldsymbol{w}}^{\mathcal{S}_k}) \mid x_i \in \mathcal{C}_{(j,x_0,r)} \right]$, *where* $S_{(j,x_0,r)} = \{(x,y) \in S_j : x \in \mathcal{B}(x_0, r), (x_0, y_0) \in S\}$.

We first prove the lemma 1 and use it to prove theorem 1, and we need lemma 2 and 3 before proving the lemma 1.

**Lemma 1** *Given* $\mathcal{B}(x_0, r)$, $(x_0, y_0) \in S_j$, $S_{(j,x_0,r)}$, *and* $f_{\boldsymbol{w}}^{\mathcal{S}_k} \in \mathcal{F}_{\boldsymbol{w}}^{\mathcal{S}_k}$, *then with probability at least* $1 - \delta$, *the following holds for any* $f_{\boldsymbol{w}}^{\mathcal{S}_k} \in \mathcal{F}_{\boldsymbol{w}}^{\mathcal{S}_k}$ *and* $k \in [K]$:

$$\left| \mathbb{E}_{x \sim \mathcal{D}_{x|y}} \left[ f_{\boldsymbol{w}}^{\mathcal{S}_k}(x) \mid x \in \mathcal{C}_{(j,x_0,r)} \right] - \hat{\mathbb{E}}_{S_{(j,x_0,r)}} \left[ f_{\boldsymbol{w}}^{\mathcal{S}_k} \right] \right|$$

$$\leq 4C^{\boldsymbol{w}} \sqrt{\frac{K}{\left| \mathcal{I}_{(j,x_0,r)} \right|}} \log^2 \left( \frac{\sqrt{2 \left| \mathcal{I}_{(j,x_0,r)} \right|}}{b^{\boldsymbol{w}}} \right) + 3 \sqrt{\frac{\log(2/\delta)}{2 \left| \mathcal{I}_{(j,x_0,r)} \right|}},$$

*where* $K$ *is the number of classes,* $C^{\boldsymbol{w}}$ *and* $b^{\boldsymbol{w}}$ *are constants.*

**Lemma 2** ($l_\infty$ **Contraction Inequality [54]**) *Let* $\mathcal{F} \subseteq \{f : \mathcal{X} \to \mathbb{R}^K\}$, *and let* $\phi : \mathbb{R}^K \to \mathbb{R}$ *be* $L$-*Lipschitz with respect to the* $l_\infty$ *norm, i.e.* $\|\phi(v) - \phi(v')\|_\infty \leq L \cdot \|v - v'\|_\infty, \forall v, v' \in \mathbb{R}^K$. *For any* $a > 0$, *there exists a constant* $C > 0$ *such that if* $|\phi(f(x))| \vee \|f(x)\|_\infty \leq \zeta$, *then*

$$\mathcal{R}_n(\phi \circ \mathcal{F}) \leq C \cdot L \sqrt{K} \max_i \tilde{\mathcal{R}}_n (\mathcal{F}_i) \log^{\frac{3}{2} + a} \left( \frac{\zeta n}{\max_i \tilde{\mathcal{R}}_n (\mathcal{F}_i)} \right),$$

*where* $\mathcal{R}_n(\phi \circ \mathcal{F}) = \mathbb{E}_{\boldsymbol{\sigma}} \left[ \sup_{f \in \mathcal{F}} |\sum_{i=1}^n \sigma_i \phi (f(\mathbf{x}_i))| \right]$, $\tilde{\mathcal{R}}_n (\mathcal{F}_i) = \sup_{S \in \mathcal{X}^n} \mathcal{R}_n (\mathcal{F}_i)$.

**Lemma 3** *Given* $\mathcal{B}(x_0, r)$, $(x_0, y_0) \in S_j$, $S_{(j,x_0,r)}$ *and* $\mathcal{F}_{\boldsymbol{w}}^{\mathcal{S}_k}$, *with probability at least* $1 - \delta$, *holds:*

$$\mathcal{R}^{(j,x_0,r)}(\mathcal{F}_{\boldsymbol{w}}^{\mathcal{S}_k}) \leq \mathcal{R}_n^{(j,x_0,r)}(\mathcal{F}_{\boldsymbol{w}}^{\mathcal{S}_k}) + \sqrt{\frac{\left| \mathcal{I}_{(j,x_0,r)} \right| \log(1/\delta)}{2}}.$$

*Proof of Lemma 3* Let $S'_{(j,x_0,r)}$ be a dataset that has at most one element different with $S_{(j,x_0,r)}$, its elements are $x'_i, i \in \mathcal{I}_{(j,x_0,r)}$, then

$$\mathbb{E}_{\boldsymbol{\sigma}}\left[\sup_{f_{\boldsymbol{w}}^{\mathcal{S}_k}\in\mathcal{F}_{\boldsymbol{w}}^{\mathcal{S}_k}}\left|\sum_{i\in\mathcal{I}_{(j,x_0,r)}}\sigma_i f_{\boldsymbol{w}}^{\mathcal{S}_k}(x_i)\right|\right]-\mathbb{E}_{\boldsymbol{\sigma}}\left[\sup_{f_{\boldsymbol{w}}^{\mathcal{S}_k}\in\mathcal{F}_{\boldsymbol{w}}^{\mathcal{S}_k}}\left|\sum_{i\in\mathcal{I}_{(j,x_0,r)}}\sigma_i f_{\boldsymbol{w}}^{\mathcal{S}_k}(x_i')\right|\right]$$

$$\leq\mathbb{E}_{\boldsymbol{\sigma}}\left[\sup_{f_{\boldsymbol{w}}^{\mathcal{S}_k}\in\mathcal{F}_{\boldsymbol{w}}^{\mathcal{S}_k}}\left[\left|\sum_{i\in\mathcal{I}_{(j,x_0,r)}}\sigma_i f_{\boldsymbol{w}}^{\mathcal{S}_k}(x_i)\right|-\left|\sum_{i\in\mathcal{I}_{(j,x_0,r)}}\sigma_i f_{\boldsymbol{w}}^{\mathcal{S}_k}(x_i')\right|\right]\right]$$

$$\leq\mathbb{E}_{\boldsymbol{\sigma}}\left[\sup_{f_{\boldsymbol{w}}^{\mathcal{S}_k}\in\mathcal{F}_{\boldsymbol{w}}^{\mathcal{S}_k}}\left|\sum_{i\in\mathcal{I}_{(j,x_0,r)}}\sigma_i\left(f_{\boldsymbol{w}}^{\mathcal{S}_k}(x_i)-f_{\boldsymbol{w}}^{\mathcal{S}_k}(x_i')\right)\right|\right]\leq 1,$$

the last inequality is because $S'_{(j,x_0,r)}$ differs from $S_{(j,x_0,r)}$ with at most one element, and $f_{\boldsymbol{w}}^{\mathcal{S}_k}\in[0,1]$. Then use McDiarmid's inequality ([55], Theorem D.3), we have

$$\mathcal{R}^{(j,x_0,r)}(\mathcal{F}_{\boldsymbol{w}}^{\mathcal{S}_k})\leq\mathcal{R}_n^{(j,x_0,r)}(\mathcal{F}_{\boldsymbol{w}}^{\mathcal{S}_k})+\sqrt{\frac{\left|\mathcal{I}_{(j,x_0,r)}\right|\log(1/\delta)}{2}}$$

holds with probability at least $1-\delta$.

In the following we use the notation: $\hat{\mathbb{E}}_{S_{(j,x_0,r)}}\left[f_{\boldsymbol{w}}^{\mathcal{S}_k}\right]=\frac{1}{\left|\mathcal{I}_{(j,x_0,r)}\right|}\sum_{i\in\mathcal{I}_{(j,x_0,r)}}f_{\boldsymbol{w}}^{\mathcal{S}_k}(x_i)$, then define:

$$\Phi(S_{(j,x_0,r)})=\sup_{f_{\boldsymbol{w}}^{\mathcal{S}_k}\in\mathcal{F}_{\boldsymbol{w}}^{\mathcal{S}_k}}\left|\mathbb{E}_{x\sim\mathcal{D}_{x|y}}\left[f_{\boldsymbol{w}}^{\mathcal{S}_k}(x)\mid x\in\mathcal{C}_{(j,x_0,r)}\right]-\hat{\mathbb{E}}_{S_{(j,x_0,r)}}\left[f_{\boldsymbol{w}}^{\mathcal{S}_k}\right]\right|.$$

**Lemma 4** *Given $\mathcal{B}(x_0,r)$, $(x_0,y_0)\in S_j$ and $f_{\boldsymbol{w}}^{\mathcal{S}_k}\in\mathcal{F}_{\boldsymbol{w}}^{\mathcal{S}_k}$, with probability at least $1-\delta$, the following holds for any $f_{\boldsymbol{w}}^{\mathcal{S}_k}\in\mathcal{F}_{\boldsymbol{w}}^{\mathcal{S}_k}$:*

$$\left|\mathbb{E}_{x\sim\mathcal{D}_{x|y}}\left[f_{\boldsymbol{w}}^{\mathcal{S}_k}(x)\mid x\in\mathcal{C}_{(j,x_0,r)}\right]-\hat{\mathbb{E}}_{S_{(j,x_0,r)}}\left[f_{\boldsymbol{w}}^{\mathcal{S}_k}\right]\right|\leq\frac{2}{\left|\mathcal{I}_{(j,x_0,r)}\right|}\mathcal{R}^{(j,x_0,r)}(\mathcal{F}_{\boldsymbol{w}}^{\mathcal{S}_k})+\sqrt{\frac{\log(1/\delta)}{2\left|\mathcal{I}_{(j,x_0,r)}\right|}}$$

*Proof of Lemma 4*  Similarly, let $S'_{(j,x_0,r)}$ be a dataset that has at most one element different with $S_{(j,x_0,r)}$, then we have

$$\Phi(S'_{(j,x_0,r)})-\Phi(S_{(j,x_0,r)})=\sup_{f_{\boldsymbol{w}}^{\mathcal{S}_k}\in\mathcal{F}_{\boldsymbol{w}}^{\mathcal{S}_k}}\left|\mathbb{E}_{x\sim\mathcal{D}_{x|y}}\left[f_{\boldsymbol{w}}^{\mathcal{S}_k}(x)\mid x\in\mathcal{C}_{(j,x_0,r)}\right]-\hat{\mathbb{E}}_{S'_{(j,x_0,r)}}\left[f_{\boldsymbol{w}}^{\mathcal{S}_k}\right]\right|$$

$$-\sup_{f_{\boldsymbol{w}}^{\mathcal{S}_k}\in\mathcal{F}_{\boldsymbol{w}}^{\mathcal{S}_k}}\left|\mathbb{E}_{x\sim\mathcal{D}_{x|y}}\left[f_{\boldsymbol{w}}^{\mathcal{S}_k}(x)\mid x\in\mathcal{C}_{(j,x_0,r)}\right]-\hat{\mathbb{E}}_{S_{(j,x_0,r)}}\left[f_{\boldsymbol{w}}^{\mathcal{S}_k}\right]\right|$$

$$\leq\sup_{f_{\boldsymbol{w}}^{\mathcal{S}_k}\in\mathcal{F}_{\boldsymbol{w}}^{\mathcal{S}_k}}\left|\hat{\mathbb{E}}_{S'_{(j,x_0,r)}}\left[f_{\boldsymbol{w}}^{\mathcal{S}_k}\right]-\hat{\mathbb{E}}_{S_{(j,x_0,r)}}\left[f_{\boldsymbol{w}}^{\mathcal{S}_k}\right]\right|$$

$$=\sup_{f_{\boldsymbol{w}}^{\mathcal{S}_k}\in\mathcal{F}_{\boldsymbol{w}}^{\mathcal{S}_k}}\left|\frac{1}{\left|\mathcal{I}_{(j,x_0,r)}\right|}\sum_{i\in\mathcal{I}_{(j,x_0,r)}}\left(f_{\boldsymbol{w}}^{\mathcal{S}_k}(x_i')-f_{\boldsymbol{w}}^{\mathcal{S}_k}(x_i)\right)\right|$$

$$\leq\frac{1}{\left|\mathcal{I}_{(j,x_0,r)}\right|},$$

then use McDiarmid's inequality, with probability at least $1-\delta$, the following holds:

$$\Phi(S_{(j,x_0,r)})\leq\mathbb{E}_{S_{(j,x_0,r)}}\left[\Phi(S_{(j,x_0,r)})\mid x_i\in\mathcal{C}_{(j,x_0,r)}\right]+\sqrt{\frac{\log(1/\delta)}{2\left|\mathcal{I}_{(j,x_0,r)}\right|}},\tag{1}$$

then we could use the standard symmetrization technique ([56], Theorem 3.1) to derive the following:

$$\mathbb{E}_{S_{(j,x_0,r)}}\left[\Phi(S_{(j,x_0,r)}) \mid x_i \in \mathcal{C}_{(j,x_0,r)}\right]$$

$$= \mathbb{E}_{S_{(j,x_0,r)}}\left[\sup_{f_{\boldsymbol{w}}^{\mathcal{S}_k} \in \mathcal{F}_{\boldsymbol{w}}^{\mathcal{S}_k}} \left|\mathbb{E}_{x \sim \mathcal{D}_{x|y}}\left[f_{\boldsymbol{w}}^{\mathcal{S}_k}(x) \mid x \in \mathcal{C}_{(j,x_0,r)}\right] - \hat{\mathbb{E}}_{S_{(j,x_0,r)}}\left[f_{\boldsymbol{w}}^{\mathcal{S}_k}\right]\right| \mid x_i \in \mathcal{C}_{(j,x_0,r)}\right]$$

$$= \mathbb{E}_{S_{(j,x_0,r)}}\left[\sup_{f_{\boldsymbol{w}}^{\mathcal{S}_k} \in \mathcal{F}_{\boldsymbol{w}}^{\mathcal{S}_k}} \left|\mathbb{E}_{S'_{(j,x_0,r)}}\left[\hat{\mathbb{E}}_{S'_{(j,x_0,r)}}\left[f_{\boldsymbol{w}}^{\mathcal{S}_k}\right] - \hat{\mathbb{E}}_{S_{(j,x_0,r)}}\left[f_{\boldsymbol{w}}^{\mathcal{S}_k}\right] \mid x'_i \in \mathcal{C}_{(j,x_0,r)}\right]\right| \mid x_i \in \mathcal{C}_{(j,x_0,r)}\right]$$

$$\underset{Jensen}{\leq} \mathbb{E}_{S_{(j,x_0,r)},S'_{(j,x_0,r)}}\left[\sup_{f_{\boldsymbol{w}}^{\mathcal{S}_k} \in \mathcal{F}_{\boldsymbol{w}}^{\mathcal{S}_k}} \left|\hat{\mathbb{E}}_{S'_{(j,x_0,r)}}\left[f_{\boldsymbol{w}}^{\mathcal{S}_k}\right] - \hat{\mathbb{E}}_{S_{(j,x_0,r)}}\left[f_{\boldsymbol{w}}^{\mathcal{S}_k}\right]\right| \mid x_i, x'_i \in \mathcal{C}_{(j,x_0,r)}\right]$$

$$= \mathbb{E}_{S_{(j,x_0,r)},S'_{(j,x_0,r)}}\left[\sup_{f_{\boldsymbol{w}}^{\mathcal{S}_k} \in \mathcal{F}_{\boldsymbol{w}}^{\mathcal{S}_k}} \frac{1}{|\mathcal{I}_{(j,x_0,r)}|}\left|\sum_{i \in \mathcal{I}_{(j,x_0,r)}} \left(f_{\boldsymbol{w}}^{\mathcal{S}_k}(x'_i) - f_{\boldsymbol{w}}^{\mathcal{S}_k}(x_i)\right)\right| \mid x_i, x'_i \in \mathcal{C}_{(j,x_0,r)}\right]$$

$$= \mathbb{E}_{S_{(j,x_0,r)},S'_{(j,x_0,r)}}\left[\sup_{f_{\boldsymbol{w}}^{\mathcal{S}_k} \in \mathcal{F}_{\boldsymbol{w}}^{\mathcal{S}_k}} \frac{1}{|\mathcal{I}_{(j,x_0,r)}|}\left|\mathbb{E}_{\boldsymbol{\sigma}}\left[\sum_{i \in \mathcal{I}_{(j,x_0,r)}} \sigma_i \left(f_{\boldsymbol{w}}^{\mathcal{S}_k}(x'_i) - f_{\boldsymbol{w}}^{\mathcal{S}_k}(x_i)\right)\right]\right| \mid x_i, x'_i \in \mathcal{C}_{(j,x_0,r)}\right]$$

(Since $S$ and $S'$ have same distribution)

$$\underset{Jensen}{\leq} \mathbb{E}_{S_{(j,x_0,r)},S'_{(j,x_0,r)},\boldsymbol{\sigma}}\left[\sup_{f_{\boldsymbol{w}}^{\mathcal{S}_k} \in \mathcal{F}_{\boldsymbol{w}}^{\mathcal{S}_k}} \frac{1}{|\mathcal{I}_{(j,x_0,r)}|}\left|\sum_{i \in \mathcal{I}_{(j,x_0,r)}} \sigma_i \left(f_{\boldsymbol{w}}^{\mathcal{S}_k}(x'_i) - f_{\boldsymbol{w}}^{\mathcal{S}_k}(x_i)\right)\right| \mid x_i, x'_i \in \mathcal{C}_{(j,x_0,r)}\right]$$

$$\leq 2\mathbb{E}_{S_{(j,x_0,r)},\boldsymbol{\sigma}}\left[\sup_{f_{\boldsymbol{w}}^{\mathcal{S}_k} \in \mathcal{F}_{\boldsymbol{w}}^{\mathcal{S}_k}} \frac{1}{|\mathcal{I}_{(j,x_0,r)}|}\left|\sum_{i \in \mathcal{I}_{(j,x_0,r)}} \sigma_i f_{\boldsymbol{w}}^{\mathcal{S}_k}(x_i)\right| \mid x_i \in \mathcal{C}_{(j,x_0,r)}\right]$$

$$= \frac{2}{|\mathcal{I}_{(j,x_0,r)}|}\mathcal{R}^{(j,x_0,r)}(\mathcal{F}_{\boldsymbol{w}}^{\mathcal{S}_k}),$$

combine with (1), we can get:

$$\left|\mathbb{E}_{x \sim \mathcal{D}_{x|y}}\left[f_{\boldsymbol{w}}^{\mathcal{S}_k}(x) \mid x \in \mathcal{C}_{(j,x_0,r)}\right] - \hat{\mathbb{E}}_{S_{(j,x_0,r)}}\left[f_{\boldsymbol{w}}^{\mathcal{S}_k}\right]\right| \leq \frac{2}{|\mathcal{I}_{(j,x_0,r)}|}\mathcal{R}^{(j,x_0,r)}(\mathcal{F}_{\boldsymbol{w}}^{\mathcal{S}_k}) + \sqrt{\frac{\log(1/\delta)}{2\,|\mathcal{I}_{(j,x_0,r)}|}}$$

holds with probability at least $1 - \delta$.

*Proof of Lemma 1*   Let $\phi_k : \mathbb{R}^K \to \mathbb{R}$ satisfies $\phi_k(v) = v_k$, where $v = (v_1, \ldots, v_k) \in \mathbb{R}^K$. Then

$$\mathcal{R}_n^{(j,x_0,r)}(\mathcal{F}_{\boldsymbol{w}}^{\mathcal{S}_k}) = \mathcal{R}_n^{(j,x_0,r)}(\phi_k \circ \mathcal{F}_{\boldsymbol{w}}^{\mathcal{S}}),$$

since $|\phi_k(v) - \phi_k(v')| = |v_k - v'_k| \leq \|v - v'\|_\infty, \forall v, v' \in \mathbb{R}^K$, i.e. $\phi_k$ is 1-Lipschitz. For $f_{\boldsymbol{w}}^{\mathcal{S}} \in \mathcal{F}_{\boldsymbol{w}}^{\mathcal{S}}$, we have $\left\|f_{\boldsymbol{w}}^{\mathcal{S}}\right\|_\infty \leq 1$, therefore, $\left|\phi_k\left(f_{\boldsymbol{w}}^{\mathcal{S}}(x)\right)\right| \vee \left\|f_{\boldsymbol{w}}^{\mathcal{S}}(x)\right\|_\infty \leq 1$, then use Lemma 2, choose $L = 1, \zeta = 1, a = \frac{1}{2}$, there exists a constant $C_k^{\boldsymbol{w}} > 0$ such that

$$\mathcal{R}_n^{(j,x_0,r)}(\phi_k \circ \mathcal{F}_{\boldsymbol{w}}^{\mathcal{S}}) \leq C_k^{\boldsymbol{w}} \cdot \sqrt{K} \max_k \tilde{\mathcal{R}}_n^{(j,x_0,r)}\left(\mathcal{F}_{\boldsymbol{w}}^{\mathcal{S}_k}\right) \log^2\left(\frac{|\mathcal{I}_{(j,x_0,r)}|}{\max_k \tilde{R}_n^{(j,x_0,r)}\left(\mathcal{F}_{\boldsymbol{w}}^{\mathcal{S}_k}\right)}\right), \quad (2)$$

where   $\tilde{\mathcal{R}}_n^{(j,x_0,r)}\left(\mathcal{F}_{\boldsymbol{w}}^{\mathcal{S}_k}\right) = \sup_{S_{(j,x_0,r)} \in \mathcal{Z}_{(j,x_0,r)}^{|\mathcal{I}_{(j,x_0,r)}|}} \mathcal{R}_n^{(j,x_0,r)}\left(\mathcal{F}_{\boldsymbol{w}}^{\mathcal{S}_k}\right)$,   and   we   denote $\left\{(x,y) \in \mathcal{Z} : x \in \mathcal{C}_{(j,x_0,r)}\right\}$ as $\mathcal{Z}_{(j,x_0,r)}$.

For $\forall S_{(j,x_0,r)} \in \mathcal{Z}_{(j,x_0,r)}^{|\mathcal{I}_{(j,x_0,r)}|}, \forall f_{\boldsymbol{w}}^{\mathcal{S}_k} \in \mathcal{F}_{\boldsymbol{w}}^{\mathcal{S}_k}$, we have

$$\mathbb{E}_{\boldsymbol{\sigma}}\left[\left\|\sum_{i \in \mathcal{I}_{(j,x_0,r)}} \sigma_i f_{\boldsymbol{w}}^{\mathcal{S}_k}(x_i)\right\|\right] = \mathbb{E}_{\boldsymbol{\sigma}}\left[\sqrt{\left\|\sum_{i \in \mathcal{I}_{(j,x_0,r)}} \sigma_i f_{\boldsymbol{w}}^{\mathcal{S}_k}(x_i)\right\|^2}\right]$$

$$\underset{Jensen}{\leq} \sqrt{\mathbb{E}_{\boldsymbol{\sigma}}\left[\left\|\sum_{i \in \mathcal{I}_{(j,x_0,r)}} \sigma_i f_{\boldsymbol{w}}^{\mathcal{S}_k}(x_i)\right\|^2\right]}$$

$$\leq \sqrt{\sum_{i \in \mathcal{I}_{(j,x_0,r)}} \left|f_{\boldsymbol{w}}^{\mathcal{S}_k}(x_i)\right|^2} \leq \sqrt{n},$$

the penultimate inequality uses the Khintchine-Kahane Inequality [57]. Therefore,

$$\mathcal{R}_n^{(j,x_0,r)}\left(\mathcal{F}_{\boldsymbol{w}}^{\mathcal{S}_k}\right) = \mathbb{E}_{\boldsymbol{\sigma}}\left[\sup_{f_{\boldsymbol{w}}^{\mathcal{S}_k} \in \mathcal{F}_{\boldsymbol{w}}^{\mathcal{S}_k}} \left|\sum_{i \in \mathcal{I}_{(j,x_0,r)}} \sigma_i f_{\boldsymbol{w}}^{\mathcal{S}_k}(x_i)\right|\right] \leq \sqrt{n},$$

which means $\tilde{\mathcal{R}}_n^{(j,x_0,r)}\left(\mathcal{F}_{\boldsymbol{w}}^{\mathcal{S}_k}\right) \leq \sqrt{n}$.

Choose $(\tilde{x}, \tilde{y}) \in \mathcal{Z}_{(j,x_0,r)}, \tilde{f}_{\boldsymbol{w}}^{\mathcal{S}_k} \in \tilde{\mathcal{F}}_{\boldsymbol{w}}^{\mathcal{S}_k}$ satisfies

$$\mathbb{E}_{\boldsymbol{\sigma}}\left[\left\|\sum_{i \in \mathcal{I}_{(j,x_0,r)}} \sigma_i \tilde{f}_{\boldsymbol{w}}^{\mathcal{S}_k}(\tilde{x})\right\|\right] = \sup_{z \in \mathcal{Z}_{(j,x_0,r)}, f_{\boldsymbol{w}}^{\mathcal{S}_k} \in \mathcal{F}_{\boldsymbol{w}}^{\mathcal{S}_k}} \mathbb{E}_{\boldsymbol{\sigma}}\left[\left\|\sum_{i \in \mathcal{I}_{(j,x_0,r)}} \sigma_i f_{\boldsymbol{w}}^{\mathcal{S}_k}(x)\right\|\right],$$

then, similar to Lemma 3 in [58], we have

$$\sup_{S_{(j,x_0,r)} \in \mathcal{Z}_{(j,x_0,r)}^{|\mathcal{I}_{(j,x_0,r)}|}} \mathcal{R}_n^{(j,x_0,r)}\left(\mathcal{F}_{\boldsymbol{w}}^{\mathcal{S}_k}\right) = \sup_{S_{(j,x_0,r)} \in \mathcal{Z}_{(j,x_0,r)}^{|\mathcal{I}_{(j,x_0,r)}|}} \mathbb{E}_{\boldsymbol{\sigma}}\left[\sup_{f_{\boldsymbol{w}}^{\mathcal{S}_k} \in \mathcal{F}_{\boldsymbol{w}}^{\mathcal{S}_k}} \left|\sum_{i \in \mathcal{I}_{(j,x_0,r)}} \sigma_i f_{\boldsymbol{w}}^{\mathcal{S}_k}(x_i)\right|\right]$$

$$\geq \sup_{z \in \mathcal{Z}_{(j,x_0,r)}} \mathbb{E}_{\boldsymbol{\sigma}}\left[\sup_{f_{\boldsymbol{w}}^{\mathcal{S}_k} \in \mathcal{F}_{\boldsymbol{w}}^{\mathcal{S}_k}} \left|\sum_{i \in \mathcal{I}_{(j,x_0,r)}} \sigma_i f_{\boldsymbol{w}}^{\mathcal{S}_k}(x)\right|\right]$$

$$\underset{Jensen}{\geq} \sup_{z \in \mathcal{Z}_{(j,x_0,r)}, f_{\boldsymbol{w}}^{\mathcal{S}_k} \in \mathcal{F}_{\boldsymbol{w}}^{\mathcal{S}_k}} \mathbb{E}_{\boldsymbol{\sigma}}\left[\left\|\sum_{i \in \mathcal{I}_{(j,x_0,r)}} \sigma_i f_{\boldsymbol{w}}^{\mathcal{S}_k}(x)\right\|\right]$$

$$= \mathbb{E}_{\boldsymbol{\sigma}}\left[\left\|\sum_{i \in \mathcal{I}_{(j,x_0,r)}} \sigma_i \tilde{f}_{\boldsymbol{w}}^{\mathcal{S}_k}(\tilde{x})\right\|\right]$$

$$\geq \frac{\sqrt{2}}{2}\sqrt{\sum_{i \in \mathcal{I}_{(j,x_0,r)}} \left|\tilde{f}_{\boldsymbol{w}}^{\mathcal{S}_k}(\tilde{x})\right|^2}$$

$$= \frac{\sqrt{2\left|\mathcal{I}_{(j,x_0,r)}\right|}}{2}\left|\tilde{f}_{\boldsymbol{w}}^{\mathcal{S}_k}(\tilde{x})\right|,$$

similarly, the penultimate inequality also uses the Khintchine-Kahane Inequality [57]. Below we denote $\left|\tilde{f}_{\boldsymbol{w}}^{\mathcal{S}_k}(\tilde{x})\right|$ as $b^{\boldsymbol{w},k}$. Choose $b^{\boldsymbol{w}} = \max_{1 \leq k \leq K} b^{\boldsymbol{w},k}$, which satisfies $\max_k \tilde{R}_n^{(j,x_0,r)} \geq \frac{\sqrt{2|\mathcal{I}_{(j,x_0,r)}|}b^{\boldsymbol{w}}}{2}$, and $C^{\boldsymbol{w}} = \max_{1 \leq k \leq K} C_k^{\boldsymbol{w}}$, combine with (2), we have

$$\mathcal{R}_n^{(j,x_0,r)}(\phi_k \circ \mathcal{F}_{\boldsymbol{w}}^{\mathcal{S}}) \leq C^{\boldsymbol{w}} \cdot \sqrt{K\left|\mathcal{I}_{(j,x_0,r)}\right|}\log^2\left(\frac{\sqrt{2\left|\mathcal{I}_{(j,x_0,r)}\right|}}{b^{\boldsymbol{w}}}\right) \tag{3}$$

holds for all $k \in [K]$. Then use Lemma 3 and Lemma 4, with probability at least $1 - \delta$, holds

$$\left| \mathbb{E}_{x \sim \mathcal{D}_{x|y}} \left[ f_{\boldsymbol{w}}^{\mathcal{S}_k}(x) \mid x \in \mathcal{C}_{(j,x_0,r)} \right] - \hat{\mathbb{E}}_{S_{(j,x_0,r)}} \left[ f_{\boldsymbol{w}}^{\mathcal{S}_k} \right] \right|$$

$$\leq \frac{2}{\left| \mathcal{I}_{(j,x_0,r)} \right|} \mathcal{R}_n^{(j,x_0,r)}(\mathcal{F}_{\boldsymbol{w}}^{\mathcal{S}_k}) + 3\sqrt{\frac{\log(2/\delta)}{2 \left| \mathcal{I}_{(j,x_0,r)} \right|}}$$

then combine with (3), we have

$$\left| \mathbb{E}_{x \sim \mathcal{D}_{x|y}} \left[ f_{\boldsymbol{w}}^{\mathcal{S}_k}(x) \mid x \in \mathcal{C}_{(j,x_0,r)} \right] - \hat{\mathbb{E}}_{S_{(j,x_0,r)}} \left[ f_{\boldsymbol{w}}^{\mathcal{S}_k} \right] \right|$$

$$\leq 4C^{\boldsymbol{w}} \sqrt{\frac{K}{\left| \mathcal{I}_{(j,x_0,r)} \right|}} \log^2 \left( \frac{\sqrt{2 \left| \mathcal{I}_{(j,x_0,r)} \right|}}{b^{\boldsymbol{w}}} \right) + 3\sqrt{\frac{\log(2/\delta)}{2 \left| \mathcal{I}_{(j,x_0,r)} \right|}}$$

holds with probability at least $1 - \delta$.

*Proof of Theroem 1* For the inequality in Lemma 1, by taking $\delta' = \frac{\delta}{K}$ for each $k \in [K]$, we can guarantee with probability at least $1 - \delta$, the inequality

$$\left| \mathbb{E}_{x \sim \mathcal{D}_{x|y}} \left[ f_{\boldsymbol{w}}^{\mathcal{S}_k}(x) \mid x \in \mathcal{C}_{(j,x_0,r)} \right] - \hat{\mathbb{E}}_{S_{(j,x_0,r)}} \left[ f_{\boldsymbol{w}}^{\mathcal{S}_k} \right] \right|$$

$$\leq 4C^{\boldsymbol{w}} \sqrt{\frac{K}{\left| \mathcal{I}_{(j,x_0,r)} \right|}} \log^2 \left( \frac{\sqrt{2 \left| \mathcal{I}_{(j,x_0,r)} \right|}}{b^{\boldsymbol{w}}} \right) + 3\sqrt{\frac{\log(2K/\delta)}{2 \left| \mathcal{I}_{(j,x_0,r)} \right|}}$$

simultaneously holds for each $k \in [K]$. Then for two different parametrized class $\mathcal{F}_{\boldsymbol{w}_1}^{\mathcal{S}_k} := \left\{ f_{\boldsymbol{w}_1}^{\mathcal{S}_k} : \boldsymbol{w}_1 \in \mathcal{W}_1 \right\}$ and $\mathcal{F}_{\boldsymbol{w}}^{\mathcal{S}_k} := \left\{ f_{\boldsymbol{w}}^{\mathcal{S}_k} : \boldsymbol{w} \in \mathcal{W} \right\}$, just take $C = \max \left\{ C^{\boldsymbol{w}_1}, C^{\boldsymbol{w}} \right\}$ and $b = \min \left\{ b^{\boldsymbol{w}_1}, b^{\boldsymbol{w}} \right\}$, the following simultaneously holds for any $i \in \{1, 2\}$ and $k \in [K]$:

$$\left| \mathbb{E}_{x \sim \mathcal{D}_{x|y}} \left[ f_{\boldsymbol{w}_i}^{\mathcal{S}_k}(x) \mid x \in \mathcal{C}_{(j,x_0,r)} \right] - \hat{\mathbb{E}}_{S_{(j,x_0,r)}} \left[ f_{\boldsymbol{w}_i}^{\mathcal{S}_k} \right] \right|$$

$$\leq 4C \sqrt{\frac{K}{\left| \mathcal{I}_{(j,x_0,r)} \right|}} \log^2 \left( \frac{\sqrt{2 \left| \mathcal{I}_{(j,x_0,r)} \right|}}{b} \right) + 3\sqrt{\frac{\log(2K/\delta)}{2 \left| \mathcal{I}_{(j,x_0,r)} \right|}}.$$

According to the assumption $\frac{1}{\left| \mathcal{I}_{(j,x_0,r)} \right|} \left| \sum_{i \in \mathcal{I}_{(j,x_0,r)}} \left( f_{\boldsymbol{w}_1}^{\mathcal{S}_k}(x_i) - f_{\boldsymbol{w}}^{\mathcal{S}_k}(x_i) \right) \right| \leq \gamma$, which means $\left| \hat{\mathbb{E}}_{S_{(j,x_0,r)}} \left[ f_{\boldsymbol{w}_1}^{\mathcal{S}_k} \right] - \hat{\mathbb{E}}_{S_{(j,x_0,r)}} \left[ f_{\boldsymbol{w}}^{\mathcal{S}_k} \right] \right| \leq \gamma$, then we have

$$\left| \mathbb{E}_{x \sim \mathcal{D}_{x|y}} \left[ f_{\boldsymbol{w}_1}^{\mathcal{S}_k}(x) \mid x \in \mathcal{C}_{(j,x_0,r)} \right] - \mathbb{E}_{x \sim \mathcal{D}_{x|y}} \left[ f_{\boldsymbol{w}}^{\mathcal{S}_k}(x) \mid x \in \mathcal{C}_{(j,x_0,r)} \right] \right|$$

$$\leq \left| \mathbb{E}_{x \sim \mathcal{D}_{x|y}} \left[ f_{\boldsymbol{w}_1}^{\mathcal{S}_k}(x) \mid x \in \mathcal{C}_{(j,x_0,r)} \right] - \hat{\mathbb{E}}_{S_{(j,x_0,r)}} \left[ f_{\boldsymbol{w}_1}^{\mathcal{S}_k} \right] \right|$$

$$+ \left| \hat{\mathbb{E}}_{S_{(j,x_0,r)}} \left[ f_{\boldsymbol{w}_1}^{\mathcal{S}_k} \right] - \hat{\mathbb{E}}_{S_{(j,x_0,r)}} \left[ f_{\boldsymbol{w}}^{\mathcal{S}_k} \right] \right|$$

$$+ \left| \mathbb{E}_{x \sim \mathcal{D}_{x|y}} \left[ f_{\boldsymbol{w}}^{\mathcal{S}_k}(x) \mid x \in \mathcal{C}_{(j,x_0,r)} \right] - \hat{\mathbb{E}}_{S_{(j,x_0,r)}} \left[ f_{\boldsymbol{w}}^{\mathcal{S}_k} \right] \right|$$

$$\leq 8C \sqrt{\frac{K}{\left| \mathcal{I}_{(j,x_0,r)} \right|}} \log^2 \left( \frac{\sqrt{2 \left| \mathcal{I}_{(j,x_0,r)} \right|}}{b} \right) + 6\sqrt{\frac{\log(2K/\delta)}{2 \left| \mathcal{I}_{(j,x_0,r)} \right|}} + \gamma,$$

thus,

$$\left\| \mathbb{E}_{x \sim \mathcal{D}_{x|y}} \left[ f_{\boldsymbol{w}_1}^{\mathcal{S}}(x) \mid x \in \mathcal{C}_{(j,x_0,r)} \right] - \mathbb{E}_{x \sim \mathcal{D}_{x|y}} \left[ f_{\boldsymbol{w}}^{\mathcal{S}}(x) \mid x \in \mathcal{C}_{(j,x_0,r)} \right] \right\|_{\infty}$$

$$= \max_{k \in [K]} \left| \mathbb{E}_{x \sim \mathcal{D}_{x|y}} \left[ f_{\boldsymbol{w}_1}^{\mathcal{S}_k}(x) \mid x \in \mathcal{C}_{(j,x_0,r)} \right] - \mathbb{E}_{x \sim \mathcal{D}_{x|y}} \left[ f_{\boldsymbol{w}}^{\mathcal{S}_k}(x) \mid x \in \mathcal{C}_{(j,x_0,r)} \right] \right|$$

$$\leq 8C \sqrt{\frac{K}{\left| \mathcal{I}_{(j,x_0,r)} \right|}} \log^2 \left( \frac{\sqrt{2 \left| \mathcal{I}_{(j,x_0,r)} \right|}}{b} \right) + 6\sqrt{\frac{\log(2K/\delta)}{2 \left| \mathcal{I}_{(j,x_0,r)} \right|}} + \gamma.$$

By the definition, we have $\left| \mathcal{I}_{(j,x_0,r)} \right| = \rho_{(j,x_0,r)} \cdot \text{vol} \mathcal{B}(x_0, r)$, and note that

$$\left(\frac{2r}{\sqrt{d}}\right)^d \leq \text{vol}\mathcal{B}(x_0, r) = \frac{\pi^{\frac{d}{2}}}{\Gamma(\frac{d}{2}+1)}r^d \leq 6r^d,$$

therefore we can derive that with probability at least $1 - \delta$, the following holds:

$$\left\|\mathbb{E}_{x \sim \mathcal{D}_{x|y}}\left[f_{\boldsymbol{w}_1}^{\mathcal{S}}(x) \mid x \in \mathcal{C}_{(j,x_0,r)}\right] - \mathbb{E}_{x \sim \mathcal{D}_{x|y}}\left[f_{\boldsymbol{w}}^{\mathcal{S}}(x) \mid x \in \mathcal{C}_{(j,x_0,r)}\right]\right\|_\infty$$

$$\leq 8C\sqrt{\frac{Kd^{d/2}}{\rho_{(j,x_0,r)}2^d r^d}}\log^2\left(\frac{2r^d\sqrt{3\rho_j(x_0,r)}}{b}\right) + 6\sqrt{\frac{d^{d/2}\log(2K/\delta)}{\rho_{(j,x_0,r)}2^{d+1}r^d}} + \gamma,$$

which completes the proof.

### A.2 Proof of Theorem 2

First, note that

$$\left\|R_{(j,x_0,r)}(\boldsymbol{w}_1) - R_{(j,x_0,r)}(\boldsymbol{w}_2)\right\| = \left\|\frac{1}{|\mathcal{I}_{(j,x_0,r)}|}\sum_{i \in \mathcal{I}_{(j,x_0,r)}}\left(\ell(\boldsymbol{w}_1, x_i) - \ell(\boldsymbol{w}_2, x_i)\right)\right\|$$

$$\leq \frac{1}{|\mathcal{I}_{(j,x_0,r)}|}\sum_{i \in \mathcal{I}_{(j,x_0,r)}}\left\|\ell(\boldsymbol{w}_1, x_i) - \ell(\boldsymbol{w}_2, x_i)\right\|$$

$$\leq L_1\left\|\boldsymbol{w}_1 - \boldsymbol{w}_2\right\|,$$

the second inequality holds because of Assumption 1. Then due to the Lipschitz continuous gradient of $R_{(j,x_0,r)}(\boldsymbol{w})$, we have

$$R_{(j,x_0,r)}(\boldsymbol{w}^{t+1}) - R_{(j,x_0,r)}(\boldsymbol{w}^t) \leq \left\langle\nabla_{\boldsymbol{w}}R_{(j,x_0,r)}(\boldsymbol{w}^t), \boldsymbol{w}^{t+1} - \boldsymbol{w}^t\right\rangle + \frac{L_1}{2}\left\|\boldsymbol{w}^{t+1} - \boldsymbol{w}^t\right\|^2$$

$$= -\eta_t\left\langle\nabla_{\boldsymbol{w}}R_{(j,x_0,r)}(\boldsymbol{w}^t), \frac{1}{M}\sum_{i \in \mathcal{I}_{B^t}}\nabla_{\boldsymbol{w}}\ell(\boldsymbol{w}^t, x_i)\right\rangle$$

$$+ \frac{L_1\eta_t^2}{2}\left\|\frac{1}{M}\sum_{i \in \mathcal{I}_{B^t}}\nabla_{\boldsymbol{w}}\ell(\boldsymbol{w}^t, x_i)\right\|^2$$

$$\leq -\eta_t\left\langle\nabla_{\boldsymbol{w}}R_{(j,x_0,r)}(\boldsymbol{w}^t), \frac{1}{M}\sum_{i \in \mathcal{I}_{B^t}}\nabla_{\boldsymbol{w}}\ell(\boldsymbol{w}^t, x_i)\right\rangle + \frac{L_1\eta_t^2 G^2}{2},$$

the inequality holds because of Assumption 2, prescribe notation $\xi_t = \left|\mathcal{I}_{B^t} \cap \mathcal{I}_{(j,x_0,r)}\right|$, combined with Assumption 3, 4, we have

$$-\eta_t\left\langle\nabla_{\boldsymbol{w}}R_{(j,x_0,r)}(\boldsymbol{w}^t), \frac{1}{M}\sum_{i \in \mathcal{I}_{B^t}}\nabla_{\boldsymbol{w}}\ell(\boldsymbol{w}^t, x_i)\right\rangle$$

$$= -\frac{\eta_t}{M}\left\langle\nabla_{\boldsymbol{w}}R_{(j,x_0,r)}(\boldsymbol{w}^t), \sum_{i \in \mathcal{I}_{B^t} \cap \mathcal{I}_{(j,x_0,r)}}\nabla_{\boldsymbol{w}}\ell(\boldsymbol{w}^t, x_i)\right\rangle$$

$$- \frac{\eta_t}{M}\left\langle\nabla_{\boldsymbol{w}}R_{(j,x_0,r)}(\boldsymbol{w}^t), \sum_{i \in \mathcal{I}_{B^t} \setminus \mathcal{I}_{(j,x_0,r)}}\nabla_{\boldsymbol{w}}\ell(\boldsymbol{w}^t, x_i)\right\rangle$$

$$\leq -\frac{\beta\eta_t\xi_t}{M}\left\|\nabla_{\boldsymbol{w}}R_{(j,x_0,r)}(\boldsymbol{w}^t)\right\|^2 + \frac{(M - \xi_t)\eta_t G^2}{M}$$

$$\leq -\frac{2\beta\mu\eta_t\xi_t}{M}\left(R_{(j,x_0,r)}(\boldsymbol{w}^t) - R_{(j,x_0,r)}^*\right) + \eta_t G^2,$$

choose $\eta_t = \frac{1}{\beta\mu t}$, due to $T \leq \frac{L_1}{2\beta\mu}$, $\eta_t G^2 \leq \frac{L_1\eta_t^2 G^2}{2}$.

Note that $\xi_t \sim$ Hypergeometric $\left(n, \left|\mathcal{I}_{(j,x_0,r)}\right|, M\right)$, by Hoeffding's inequality [59], we have

$$\mathbb{P}\left(\xi_t - \left|\mathcal{I}_{(j,x_0,r)}\right| \leq -M\varepsilon\right) \leq e^{-2M\varepsilon^2},$$

take $\frac{\delta}{T} = e^{-2M\varepsilon^2}$, we have

$$\frac{\xi_t}{M} \geq \frac{\left|\mathcal{I}_{(j,x_0,r)}\right|}{M} - \sqrt{\frac{\ln\left(T/\delta\right)}{2M}}$$

holds simultaneously for all $t \in \{1, 2, \ldots, T\}$ with probability at least $1 - \delta$, which means

$$- \frac{2\beta\mu\eta_t\xi_t}{M}\left(R_{(j,x_0,r)}(\boldsymbol{w}^t) - R^*_{(j,x_0,r)}\right)$$
$$\leq -2\beta\mu\eta_t \max\left\{\left(\frac{\left|\mathcal{I}_{(j,x_0,r)}\right|}{M} - \sqrt{\frac{\ln\left(T/\delta\right)}{2M}}\right), 0\right\}\left(R_{(j,x_0,r)}(\boldsymbol{w}^t) - R^*_{(j,x_0,r)}\right).$$

Therefore,

$$R_{(j,x_0,r)}(\boldsymbol{w}^{t+1}) - R^*_{(j,x_0,r)}$$
$$\leq R_{(j,x_0,r)}(\boldsymbol{w}^t) - R^*_{(j,x_0,r)}$$
$$\leq \left(1 - \frac{2}{t}\max\left\{\left(\frac{\left|\mathcal{I}_{(j,x_0,r)}\right|}{M} - \sqrt{\frac{\ln\left(T/\delta\right)}{2M}}\right), 0\right\}\right)\left(R_{(j,x_0,r)}(\boldsymbol{w}^t) - R^*_{(j,x_0,r)}\right) + \frac{L_1 G^2}{2\beta^2\mu^2 t^2}$$
$$= \left(1 - \frac{2}{t}\max\left\{\left(\frac{\rho_{(j,x_0,r)}\pi^{d/2}r^d}{\Gamma\left(\frac{d}{2}+1\right)M} - \sqrt{\frac{\ln\left(T/\delta\right)}{2M}}\right), 0\right\}\right)\left(R_{(j,x_0,r)}(\boldsymbol{w}^t) - R^*_{(j,x_0,r)}\right) + \frac{L_1 G^2}{2\beta^2\mu^2 t^2}$$

holds simultaneously for all $t \in \{1, 2, \ldots, T\}$ with probability at least $1 - \delta$.

### A.3 Proof of Proposition 1

Under Assumption 1, due to the Lipschitz continuous gradient, for $\forall(x_i, y_i) \in S$, we have

$$\ell\left(\boldsymbol{w}, x_i\right) \geq \ell\left(\boldsymbol{w}, x\right) + \left\langle\nabla_x\ell\left(\boldsymbol{w}, x\right), x_i - x\right\rangle - \frac{L_2}{2}\left\|x_i - x\right\|^2, \tag{4}$$

$$\ell\left(\boldsymbol{w}, x_i\right) \leq \ell\left(\boldsymbol{w}, x\right) + \left\langle\nabla_x\ell\left(\boldsymbol{w}, x\right), x_i - x\right\rangle + \frac{L_2}{2}\left\|x_i - x\right\|^2 \tag{5}$$

$$\ell\left(\boldsymbol{w}, x\right) \leq \ell\left(\boldsymbol{w}, x_i\right) + \left\langle\nabla_x\ell\left(\boldsymbol{w}, x_i\right), x - x_i\right\rangle + \frac{L_2}{2}\left\|x_i - x\right\|^2 \tag{6}$$

holds for $\forall x \in \mathcal{X}$. By (4), choose $x = x_i - r\frac{\nabla_x\ell(\boldsymbol{w},x)}{\|\nabla_x\ell(\boldsymbol{w},x)\|}$, which satisfies $\|x_i - x\| \leq r$, then we have

$$\ell\left(\boldsymbol{w}, x_i\right) \geq \ell\left(\boldsymbol{w}, x_i - r\frac{\nabla_x\ell\left(\boldsymbol{w}, x\right)}{\|\nabla_x\ell\left(\boldsymbol{w}, x\right)\|}\right) + r\left\|\nabla_x\ell\left(\boldsymbol{w}, x\right)\right\| - \frac{L_2 r^2}{2}$$

$$\geq r\left\|\nabla_x\ell\left(\boldsymbol{w}, x\right)\right\| - \frac{L_2 r^2}{2}$$

$$= r\left\|\nabla_x\ell\left(\boldsymbol{w}, x\right) - \nabla_x\ell\left(\boldsymbol{w}, x_i\right) + \nabla_x\ell\left(\boldsymbol{w}, x_i\right)\right\| - \frac{L_2 r^2}{2}$$

$$\geq r\left(\left\|\nabla_x\ell\left(\boldsymbol{w}, x_i\right)\right\| - \left\|\nabla_x\ell\left(\boldsymbol{w}, x_i\right) - \nabla_x\ell\left(\boldsymbol{w}, x\right)\right\|\right) - \frac{L_2 r^2}{2}$$

$$\geq r\left(\left\|\nabla_x\ell\left(\boldsymbol{w}, x_i\right)\right\| - L_2\left\|x_i - x\right\|\right) - \frac{L_2 r^2}{2}$$

$$\geq r\left\|\nabla_x\ell\left(\boldsymbol{w}, x_i\right)\right\| - \frac{3L_2 r^2}{2},$$

which means

$$\|\nabla_x \ell(\boldsymbol{w}, x_i)\| \leq \frac{\ell(\boldsymbol{w}, x_i)}{r} + \frac{3L_2 r}{2}. \tag{7}$$

Use (5), we have

$$\ell(\boldsymbol{w}, x_i) - \ell(\boldsymbol{w}, x) \leq \langle \nabla_x \ell(\boldsymbol{w}, x), x_i - x \rangle + \frac{L_2}{2} \|x_i - x\|^2$$

$$\leq r \|\nabla_x \ell(\boldsymbol{w}, x)\| + \frac{L_2 r^2}{2}$$

$$\leq r (\|\nabla_x \ell(\boldsymbol{w}, x_i)\| + \|\nabla_x \ell(\boldsymbol{w}, x_i) - \nabla_x \ell(\boldsymbol{w}, x)\|) + \frac{L_2 r^2}{2}$$

$$\leq r \|\nabla_x \ell(\boldsymbol{w}, x_i)\| + \frac{3L_2 r^2}{2},$$

then use (6), we have

$$\ell(\boldsymbol{w}, x) - \ell(\boldsymbol{w}, x_i) \leq \langle \nabla_x \ell(\boldsymbol{w}, x_i), x - x_i \rangle + \frac{L_2}{2} \|x_i - x\|^2$$

$$\leq r \|\nabla_x \ell(\boldsymbol{w}, x_i)\| + \frac{L_2 r^2}{2}$$

Thus

$$\frac{|\ell(\boldsymbol{w}, x_i) - \ell(\boldsymbol{w}, x)|}{r} - \frac{3L_2 r}{2} \leq \|\nabla_x \ell(\boldsymbol{w}, x_i)\|. \tag{8}$$

Combining (7) and (8), we get

$$\frac{|\ell(\boldsymbol{w}, x_i) - \ell(\boldsymbol{w}, x)|}{r} - \frac{3L_2 r}{2} \leq \|\nabla_x \ell(\boldsymbol{w}, x_i)\| \leq \frac{\ell(\boldsymbol{w}, x_i)}{r} + \frac{3L_2 r}{2}.$$

## B  Details of Pre-training Embeddings

In actual experiments, it was found that if the embedding is not pre-trained first, the model convergence speed will be quite slow. We tried to freeze the parameters of the embedding layer after pre-training multiple epochs, but found that even with the same number of epochs each pre-training, the internal structures between the embedding vectors are very different, and there may be clustering between different class embedding (as shown in Figure6), the distance between multiple classes is very close, which leads to redundant information from many other classes in embedding.

To address this issue and enable the generator's embeddings to better reflect class information, we employ the following loss, utilizing the feature outputs of the source model as guidance for embedding pretraining:

$$\mathcal{L}_{\mathcal{M}} = \sum_{i,j} \overline{M}_{i,j}^{S_{\text{euc}}} \log \frac{\overline{M}_{i,j}^{S_{\text{euc}}}}{\overline{M}_{i,j}^{E_{\text{euc}}}} + \sum_{i,j} \overline{M}_{i,j}^{E_{\text{euc}}} \log \frac{\overline{M}_{i,j}^{E_{\text{euc}}}}{\overline{M}_{i,j}^{S_{\text{euc}}}}$$
$$+ \sum_{i,j} \overline{M}_{i,j}^{S_{\cos}} \log \frac{\overline{M}_{i,j}^{S_{\cos}}}{\overline{M}_{i,j}^{E_{\cos}}} + \sum_{i,j} \overline{M}_{i,j}^{E_{\cos}} \log \frac{\overline{M}_{i,j}}{\overline{M}_{i,j}}.$$

However, directly optimizing the $\mathcal{L}_M$ leads to the issue of loss collapse. We analyse each component of the loss. First, note that for $\forall i, j \in [K]$, $-1 \leq M_{i,j}^{E_{\cos}} \leq 1$, $-1 \leq M_{i,j}^{S_{\cos}} \leq 1$. Then

$$\overline{M}_{i,j}^{E_{\cos}} = \frac{\exp(M_{i,j}^{E_{\cos}})}{\sum_{k=1}^{K} \exp(M_{i,k}^{E_{\cos}})} = \frac{1}{1 + \sum_{k \neq j} \exp(M_{i,k}^{E_{\cos}} - M_{i,j}^{E_{\cos}})} \geq \frac{1}{1 + (K-1)\exp(2)},$$

$$\overline{M}_{i,j}^{S_{\cos}} = \frac{\exp(M_{i,j}^{S_{\cos}})}{\sum_{k=1}^{K} \exp(M_{i,k}^{S_{\cos}})} = \frac{1}{1 + \sum_{k \neq j} \exp(M_{i,k}^{S_{\cos}} - M_{i,j}^{S_{\cos}})} \geq \frac{1}{1 + (K-1)\exp(2)}.$$

Under the settings in this paper, $K = 10$, so both terms analyse above are positive with lower bounds. In the optimization process, their ratio will not approach zero. As for $\overline{M}_{i,j}^{S_{\text{euc}}}$, it is a deterministic positive constant. Therefore, when it serves as the numerator, it will not lead to a ratio close to zero inside the logarithm. However, for $\overline{M}_{i,j}^{E_{\text{euc}}}$, if it becomes close to zero when used as the numerator, it will cause loss collapse. Note that

$$\overline{M}_{i,j}^{E_{\text{euc}}} = \frac{\exp\left(M_{i,j}^{E_{\text{euc}}}\right)}{\sum_{k=1}^{K} \exp\left(M_{i,k}^{E_{\text{euc}}}\right)} = \frac{1}{1 + \sum_{k \neq j} \exp\left(M_{i,k}^{E_{\text{euc}}} - M_{i,j}^{E_{\text{euc}}}\right)} \geq \frac{1}{K \max_{1 \leq k \leq K} \exp\left(M_{i,k}^{E_{\text{euc}}} - M_{i,j}^{E_{\text{euc}}}\right)}.$$

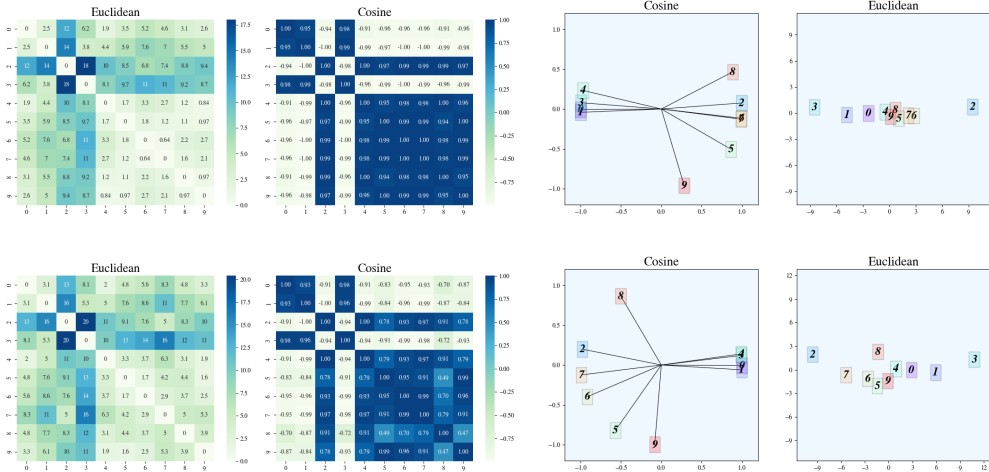

Figure 6: The top row displays the result of directly trained for 9 epochs, while the bottom row shows the result of directly trained for 49 epochs. The numbers $0 - 9$ represent different class labels in sequential order. The two heatmaps on the left depict the Euclidean distance and cosine similarity between the embeddings of each class. The two images on the right display the distances after PCA projection onto a 2-dimensional plane (normalized and unnormalized, representing cosine similarity and Euclidean distance, respectively).

To prevent $\overline{M}_{i,j}^{E_{\text{euc}}}$ from becoming too small, we need to ensure:

$$\frac{1}{K \max\limits_{1 \leq k \leq K} \exp\left(M_{i,k}^{E_{\text{euc}}} - M_{i,j}^{E_{\text{euc}}}\right)} > 0,$$

which means we need to ensure that

$$\max_{1 \leq k \leq K} \exp\left(M_{i,k}^{E_{\text{euc}}}\right) \neq +\infty, \forall 1 \leq i \leq K,$$

since $M_{i,k}^{E_{\text{euc}}}, M_{i,j}^{E_{\text{euc}}}$ are non-negative. Also, note that $M_{i,k}^{E_{\text{euc}}} = \left|e^i - e^k\right| \leq \left|e^i\right| + \left|e^k\right|$. Therefore, by imposing the following constraint in the optimization:

$$\begin{cases} \min \mathcal{L}_{\mathcal{M}} \\ s.t. \left\|e^i\right\| \leq U, 1 \leq i \leq K \end{cases}$$

We can ensure that $M_{i,k}^{E_{\text{euc}}} \leq 2U$, which in turn guarantees $\max\limits_{1 \leq k \leq K} \exp M_{i,k}^{E_{\text{euc}}} \neq +\infty, \forall 1 \leq i \leq K$, thus avoiding the issue of $\overline{M}_{i,j}^{E_{\text{euc}}}$ becoming too small and causing loss collapse.

By explicitly incorporating the above constraint as a regularizer into the optimization objective, we obtain the final manifold matching loss:

$$\mathcal{L}_{\mathcal{M}} = \sum_{i,j} \overline{M}_{i,j}^{S_{\text{euc}}} \log \frac{\overline{M}_{i,j}^{S_{\text{euc}}}}{\overline{M}_{i,j}^{E_{\text{euc}}}} + \sum_{i,j} \overline{M}_{i,j}^{E_{\text{euc}}} \log \frac{\overline{M}_{i,j}^{E_{\text{euc}}}}{\overline{M}_{i,j}^{S_{\text{euc}}}}$$
$$+ \lambda_1 \left[ \sum_{i,j} \overline{M}_{i,j}^{S_{\cos}} \log \frac{\overline{M}_{i,j}^{S_{\cos}}}{\overline{M}_{i,j}^{E_{\cos}}} + \sum_{i,j} \overline{M}_{i,j}^{E_{\cos}} \log \frac{\overline{M}_{i,j}}{\overline{M}_{i,j}} \right] + \lambda_2 \sum_{i=1}^{K} \left\|e^i\right\|.$$

In our experiments, we use the AdamW optimizer with a learning rate of $1.5e - 5$ for 15000 epochs to optimize the embeddings, while setting $\lambda_1$ to 5 and $\lambda_2$ to 0.01. The results, as shown in Figure 7, demonstrate a significant improvement in the clustering of embeddings between different classes.

The trained embedding is denoted as $e_j \in \mathbb{R}^{d_1}, j \in [K]$, where $e_j \in \mathbb{R}^{d_1}, j \in [K]$. For the utilized Resblock, let the input and output channels be $C_{in}$ and $C_{out}$ respectively. The fully connected layer

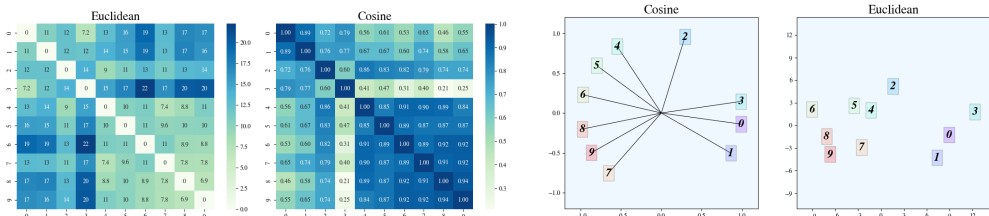

Figure 7: The results obtained by optimizing $\mathcal{L}_{\mathcal{M}}$.

$\mathrm{MLP}(\cdot) \in \mathbb{R}^{d_1} \to \mathbb{R}^{C_{out}}$. Firstly, we employ the initial convolution kernel $\mathrm{Conv}_1$ to map the raw data from the input $h$ of the current layer, transitioning it from $[C_{in}, \mathrm{H}, \mathrm{W}]$ to $[C_{out}, \mathrm{H}, \mathrm{W}]$. Then we perform an addition operation between this mapped result and the embedding transformation obtained through the MLP. Formally expressed as:

$$h' = \mathrm{Conv}_1(h) + \mathrm{MLP}(e_y)$$

where $y$ is the label of the input. Subsequently, we apply a second convolution to this output, followed by a LayerNorm operation, and finally, pass it through an attention operation before adding it to the residual. Formally stated as:

$$\mathrm{output} = \mathrm{Attention}(\mathrm{LN}(\mathrm{Conv}_2(h')) + h)$$

## C  Additional Ablation Studies

To investigate the impact of different settings during the training phase, we conduct several experiments. First, we investigate the output consistency of the model for different values of $q$ used for sample screening, as shown in Figure 8. A smaller $q$ implies a stricter selection, which leads to an increase in output consistency (in terms of mean), but also results in a larger standard deviation. In this case, $q = 5$ is a relatively suitable choice.

Then we validate the impact of different choices of $d(\cdot, \cdot)$ on targeted transferability. The results are shown in Figure 9, where $d(\cdot, \cdot)$ was chosen as the Smooth L1 loss, which significantly improves the targeted transferability compared to L1 and L2 losses.

## D  Implementation Details

In the exploratory experiments conducted in Section 2, the three different network architectures we employed are depicted in Table 4.

Table 4: Architectures of NN Classifiers

| Model | Hidden Layer 1 | Hidden Layer 2 | Hidden Layer 3 |
|---|---|---|---|
| Model 1 | Linear(2, 500)
ReLU | Linear(500, 500)
ReLU | Linear(500, 2) |
| Model 2 | Linear(2, 50)
ReLU | Linear(50, 100)
ReLU | Linear(100, 150)
ReLU
Linear(150, 2) |
| Model 3 | Linear(2, 20)
ReLU | Linear(20, 20)
ReLU | Linear(20, 20)
ReLU
Linear(20, 2) |

For the experiments on CIFAR10 mentioned in Section 2, we used the PyTorch framework. The optimizer is SGD, with an initial learning rate of $0.1$, weight decay of $5e - 4$, and momentum of $0.9$. We apply cosine annealing learning rate decay, with a maximum number of epochs ($T_{max}$) set to 200. Tolerance of Early-stopping is set to 30. Additionally, we used a batch size of 128 during training and applied the following data augmentations:

- Randomly crops the input image to a size of $32 \times 32$ pixels with padding of 4 pixels.

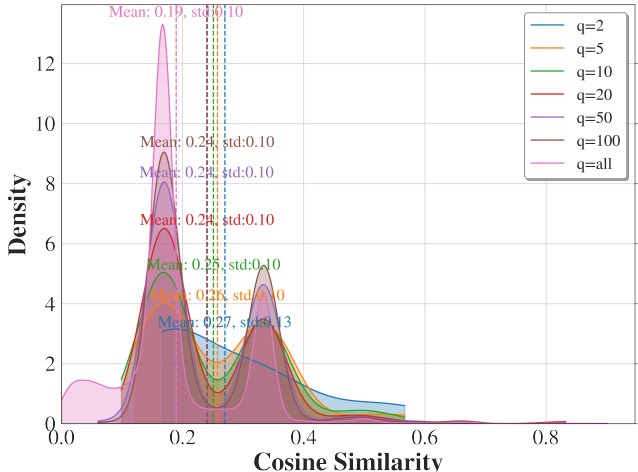

Figure 8: Density plots of output similarity between models under different $q$ selections. For the screened data, we calculate the cosine similarity between the output of the source model and the output of all target models, and take the average to represent the output similarity of the data across different models. "q=all" indicates no screening, and the dashed line represents the average for different values of $q$.

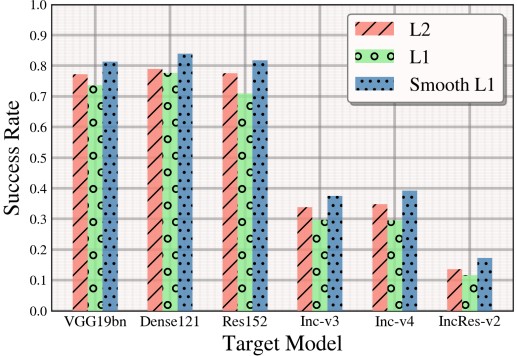

Figure 9: Targeted transferability (Src:Res50) with different chosen of $d(\cdot, \cdot)$.

• Randomly flips the input image horizontally with a probability of 0.5.

After data augmentation, the input is then transformed into a tensor and normalized with a mean of $(0.4914, 0.4822, 0.4465)$ and a standard deviation of $(0.2023, 0.1994, 0.2010)$.

For the experiments in Section 4, we used the structure of following models: ResNet-50, VGG19bn, DenseNet-121, ResNet-152 from the torchvision [1] library, and others from the timm [2] library. The torchattacks [3] library is utilized for MIM, SI-NIM, TIM, and DIM.

# E  Additional Experimental Results

## E.1  Ensemble Model as Source

In Table 5, we report the results of the case of an ensemble model as the source model. The selected ensemble model are consist of three models: Res-50, Inc-v3, and Inc-v4, while the other models

---

[1] https://github.com/pytorch/vision

[2] https://github.com/huggingface/pytorch-image-models

[3] https://github.com/Harry24k/adversarial-attacks-pytorch

not involved in the ensemble are treated as target models. ESMA still demonstrates outstanding adversarial transferability.

Table 5: Targeted transfer success rates. The source model is an ensemble of Res-50, Inc-v3, and Inc-v4.

| Attack | →VGG19bn | →Dense121 | →Res152 | →IncRes-v2 | →ViT | AVG |
|---|---|---|---|---|---|---|
| | | | Src:Ensemble | | | |
| MIM | 1.76% | 3.40% | 3.29% | 1.07% | 0.22% | 1.95% |
| TIM | 9.96% | 16.13% | 14.76% | 8.27% | 1.11% | 10.05% |
| DIM | 10.27% | 16.31% | 15.22% | 8.20% | 0.87% | 10.17% |
| SI-NIM | 1.16% | 1.67% | 1.49% | 0.93% | 0.13% | 1.08% |
| Po-TI-Trip | 14.22% | 21.76% | 18.69% | 12.09% | 1.62% | 13.68% |
| SI-FGS$^2$M | 15.72% | 24.84% | 20.79% | 12.20% | 1.53% | 15.02% |
| S$^2$I-SI-TI-DIM | 17.82% | 21.07% | 21.36% | 16.38% | 2.38% | 15.80% |
| Logit | 16.09% | 27.56% | 17.51% | 36.96% | 2.20% | 20.06% |
| DTMI-Logit-SU | 18.11% | 30.16% | 18.73% | 38.64% | 2.64% | 21.71% |
| RAP-LS | 48.98% | 72.91% | 70.67% | 26.53% | 4.22% | 44.66% |
| HGN | 55.10% | 67.37% | 63.54% | 49.85% | 26.78% | 52.53% |
| TTP(129257.2) | 60.02% | 71.33% | 67.44% | 55.42% | 30.04% | 56.85% |
| ESMA(112303.5) | **62.51%** | **74.11%** | **72.61%** | **62.33%** | **32.70%** | **60.85%** |

## E.2   Models Trained on Different Distributions as Victim

Given that our discussion is based on the condition that the source and target models are trained on the same distribution data, to explore the transferability between models trained on training data with different distribution, we conducted additional transfer attacks on two adversarially trained models, Inc-v3-adv and IncRes-v2-ens. The results are reported in Table 6. ESMA's performance is slightly weakened but still competitive. Considering that this validation is not directly related to the viewpoints, conclusions, and statements of our paper, we will further discuss the transferability on target models trained on data with distribution shift in our future work.

Table 6: Targeted transfer success rates. "Src" indicates the source model.

| Src | Target | MIM | TIM | DIM | SI-NIM | Po-TI-Trip | SI-FGS$^2$M | S$^2$I-SI-TI-DIM | Logit | DTMI-Logit-SU | RAP-LS | HGN | TTP | ESMA(Ours) |
|---|---|---|---|---|---|---|---|---|---|---|---|---|---|---|
| **Res50** | →Inc-v3$_{adv}$ | 0.11% | 0.36% | 0.18% | 0.11% | 0.22% | 0.24% | 0.76% | 0.20% | 0.27% | 0.38% | 2.12% | 3.53% | 3.23% |
| | →IncRes-v2$_{ens}$ | 0.11% | 0.22% | 0.18% | 0.02% | 0.38% | 0.30% | 0.89% | 0.18% | 0.22% | 0.27% | 2.70% | 4.53% | 3.98% |

| Src | Target | MIM | TIM | DIM | SI-NIM | Po-TI-Trip | SI-FGS$^2$M | S$^2$I-SI-TI-DIM | Logit | DTMI-Logit-SU | RAP-LS | HGN | TTP | ESMA(Ours) |
|---|---|---|---|---|---|---|---|---|---|---|---|---|---|---|
| **Dense121** | →Inc-v3$_{adv}$ | 0.13% | 0.13% | 0.18% | 0.11% | 0.18% | 0.22% | 0.56% | 0.20% | 0.20% | 0.22% | 1.88% | 3.18% | 1.73% |
| | →IncRes-v2$_{ens}$ | 0.09% | 0.18% | 0.13% | 0.11% | 0.31% | 0.20% | 0.84% | 0.22% | 0.20% | 0.18% | 2.39% | 3.56% | 1.83% |

| Src | Target | MIM | TIM | DIM | SI-NIM | Po-TI-Trip | SI-FGS$^2$M | S$^2$I-SI-TI-DIM | Logit | DTMI-Logit-SU | RAP-LS | HGN | TTP | ESMA(Ours) |
|---|---|---|---|---|---|---|---|---|---|---|---|---|---|---|
| **VGG19bn** | →Inc-v3$_{adv}$ | 0.13% | 0.11% | 0.16% | 0.11% | 0.11% | 0.22% | 0.16% | 0.13% | 0.13% | 0.18% | 0.20% | 0.49% | 0.31% |
| | → IncRes-v2$_{ens}$ | 0.04% | 0.07% | 0.09% | 0.01% | 0.11% | 0.13% | 0.22% | 0.07% | 0.07% | 0.09% | 0.53% | 0.60% | 0.22% |

| Src | Target | MIM | TIM | DIM | SI-NIM | Po-TI-Trip | SI-FGS$^2$M | S$^2$I-SI-TI-DIM | Logit | DTMI-Logit-SU | RAP-LS | HGN | TTP | ESMA(Ours) |
|---|---|---|---|---|---|---|---|---|---|---|---|---|---|---|
| **Res152** | → Inc-v3$_{adv}$ | 0.13% | 0.20% | 0.20% | 0.04% | 0.40% | 0.27% | 0.93% | 0.24% | 0.26% | 0.24% | 4.26% | 5.96% | 2.31% |
| | → IncRes-v2$_{ens}$ | 0.07% | 0.36% | 0.27% | 0.07% | 0.56% | 0.33% | 1.02% | 0.27% | 0.31% | 0.38% | 4.55% | 5.96% | 3.68% |

| Src | Target | MIM | TIM | DIM | SI-NIM | Po-TI-Trip | SI-FGS$^2$M | S$^2$I-SI-TI-DIM | Logit | DTMI-Logit-SU | RAP-LS | HGN | TTP | ESMA(Ours) |
|---|---|---|---|---|---|---|---|---|---|---|---|---|---|---|
| **Ensemble** | → Inc-v3$_{adv}$ | 0.20% | 0.60% | 0.40% | 0.13% | 0.69% | 0.60% | 2.51% | 0.53% | 0.53% | 0.58% | 22.67% | 24.44% | 17.00% |
| | → IncRes-v2$_{ens}$ | 0.11% | 0.96% | 0.42% | 0.09% | 0.87% | 0.66% | 2.51% | 0.20% | 0.24% | 0.58% | 21.25% | 23.33% | 20.82% |

# F   Fooling Google Lens

We show that adversarial examples perturbated using ESMA can deceive Google Lens in image search, as shown in Figure 10 and 11.

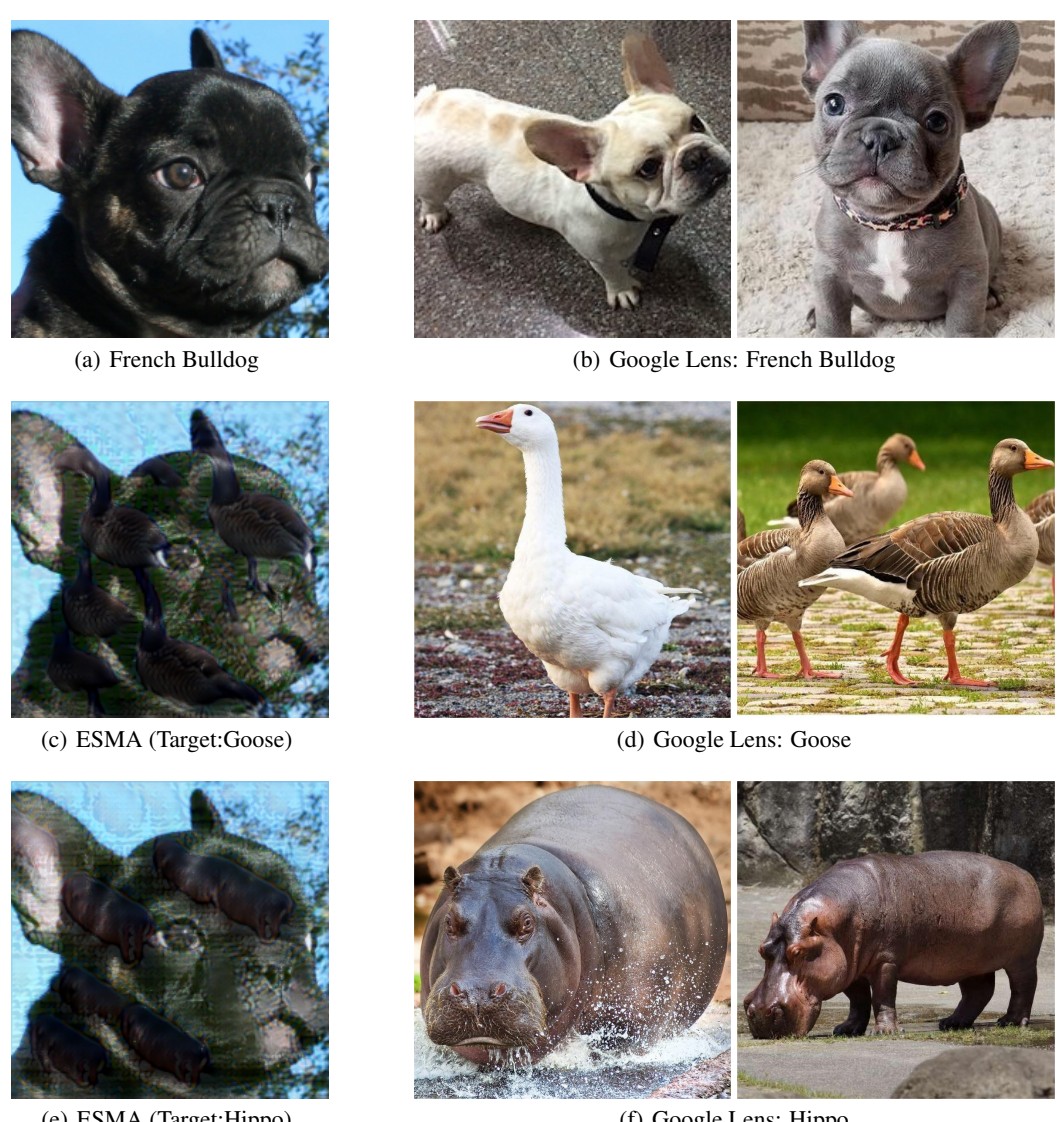

(a) French Bulldog

(b) Google Lens: French Bulldog

(c) ESMA (Target:Goose)

(d) Google Lens: Goose

(e) ESMA (Target:Hippo)

(f) Google Lens: Hippo

Figure 10: The original image is of a French Bulldog, which can be intentionally perturbed to yield different search results, such as a goose in the second row and a hippopotamus in the third row.

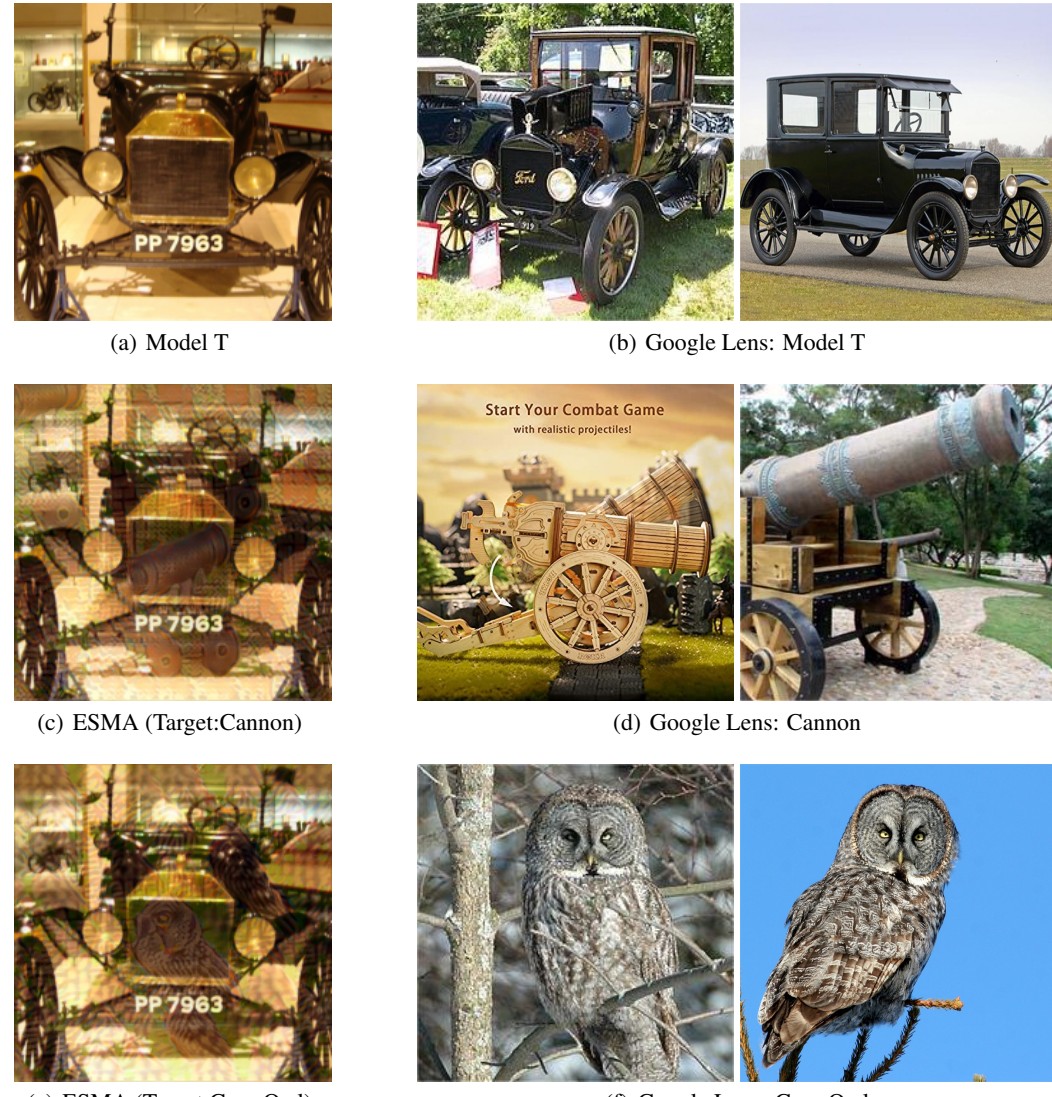

(a) Model T          (b) Google Lens: Model T

(c) ESMA (Target:Cannon)          (d) Google Lens: Cannon

(e) ESMA (Target:Grey Owl)          (f) Google Lens: Grey Owl

Figure 11: The original image is of a Ford Model T, which can be intentionally perturbed to yield different search results, such as a cannon in the second row and a gray owl in the third row.

