# OpenReview forum: "Perturbation Towards Easy Samples Improves Targeted Adversarial Transferability"
_NeurIPS.cc/2023/Conference — NeurIPS 2023 poster_

### Official Review · Reviewer_YEMS · 2023-06-10

**Soundness:** 3 good
**Presentation:** 3 good
**Contribution:** 3 good
**Rating:** 6
**Confidence:** 4

**Summary:**

The paper theoretically and experimentally demonstrated that adding perturbations towards High-Sample-Density-Regions (HSDR) of the target class can further enhance the transferability of targeted attacks. To avoided the impracticality of density estimation in high-dimensional data, authors propose a method for identifying samples within HSDR. The experimental results demonstrate the effectivenss of  the proposed method.

**Strengths:**

- The experiments demonstrate the effectiveness of the proposed ESMA, which can outperform baselines by a large margin.
- The idea of adding perturbations towards HSDR is intersting and effective.
- To avoided the impracticality of density estimation in high-dimensional data, this paper propose a method for identifying samples within HSDR, which is practical.

**Weaknesses:**

- The recent transfer-based attack [1,2,3] is missing. I suggest authors discuss or compare propsed method with these paper.
- The experiments are not extensive, i.e., only two tables. More ablation studies should be presented. For example,
   - Will the results be really poor if a difficult sample is selected to form target anchor sets? Or why author don't report the results of ESMA in Table 1.
   - Does ESMA still outperform the existing algorithm if the source model is an ensemble of models.



[1] Zhang et al. Staircase sign method for boosting adversarial attacks, 2021

[2] Zhao et al. On Success and Simplicity: A Second Look at Transferable Targeted Attacks, NeurIPS 2021

[3] Long et al. Frequency Domain Model Augmentation for Adversarial Attack, ECCV2022

**Questions:**

Please refer to weakness.

**Limitations:**

This paper only considers CNNs and it is doubtful whether it is valid on ViTs.

---

> ### Author Rebuttal · Authors · 2023-08-08
>
> Thank you very much for your valuable feedback. Below is our response to your suggestions:
>
>
> **Concern1**：The recent transfer-based attack [1,2,3] is missing. I suggest authors discuss or compare propsed method with these paper.
>
> **Response to Concern1**: We sincerely appreciate your suggestions, as they greatly contribute to improving our article. Following your advice, we have incorporated these works into the references. Regarding [1] and [2], we acknowledge their competitive nature as baseline methods and have included them in our comparative experiments (referred to as SI-FGS$^2$M and Logit, respectively, in our comparison). As for [3], we observed that it does not provide any discussion or experiments in the context of target attack scenarios; thus, we have not included it in our comparisons. However, for the comprehensiveness of the related work in the article, all mentioned papers are referenced and explained within the article.
>
> **Concern2**：Will the results be really poor if a difficult sample is selected to form target anchor sets? Or why author don't report the results of ESMA in Table 1.
>
> **Response to Concern2**: Sincerely appreciate your invaluable suggestion. In fact, we have already included comparative ablation experiments with and without anchor point screening in Appendix C of the initial draft. It is true that when randomly selecting target anchors, there is a noticeable degradation in performance. Table 1 was specifically designed to validate the broad applicability of our approach in enhancing target attack transferability. It provides a comparison between the baseline and the baseline with the additional screening mechanism, excluding ESMA. Initially, our comparison involved attacks using standard cross-entropy and attacks utilizing the square of the distance to target anchors, highlighting the improvement achieved through screening. To ensure even greater rigor, we have conscientiously included the scenario of randomly selecting anchor points in Table 1's baseline, thereby further enriching our analysis. Once again, I sincerely thank you for your input and for helping us refine our work.
>
> **Concern3**：Does ESMA still outperform the existing algorithm if the source model is an ensemble of models.
>
> **Response to Concern3**: Your questions are of immense help to us. In response to your inquiries, we utilized an ensemble model composed of ResNet-50, Inception-v3, and Inception-v4 as the source model. We then conducted tests on models outside of this ensemble, and the results of these tests have been documented in Appendix E. The experimental outcomes demonstrate that our method continues to exhibit superior performance even when the source model is an ensemble.
>
> **Concern4**：This paper only considers CNNs and it is doubtful whether it is valid on ViTs.
>
> **Response to Concern4**: Thank you sincerely for your valuable input. Your concern has truly sparked our inspiration, as the inclusion of ViT in transfer attack research is relatively uncommon at present. Taking your concern to heart, we have conscientiously incorporated ViT as one of the target models. The experimental results continue to affirm the effectiveness of our approach. We are deeply grateful for your insightful suggestion, as it has significantly contributed to the enhancement of our work.
>
> Once again, we sincerely appreciate your valuable input and suggestions. Regarding the experiments we added, we have included the tables in the supplementary PDF file attached to the author rebuttal. The experiments we added are marked in blue in the table.

---

> > ### Comment · Reviewer_YEMS · 2023-08-17
> > **Comments on the authors' rebuttal**
> >
> > Thanks for your rebuttal.
> > For concerns 2~4, thanks for the explanation and experiment and they well solve my concerns.
> > For my concern 1, I know [3] does not provide any discussion or experiments in the context of target attack scenarios, but neither do SI-NIM, TIM and DIM. I wonder the peformance of [3] in targeted attack. Could you provide this experiment?
> >
> > BTW, For algorithm 3, what are the vaules of $N$ and $n$ set to? I can't find in the paper

---

> > > ### Author Response · Authors · 2023-08-18
> > > **Response to the comment**
> > >
> > > Sincerely appreciate all your feedback and suggestions. Your input has been immensely helpful to us. Here are our responses to each of your suggestions and questions:
> > >
> > > **Response to Concern1**:
> > >
> > > Thank you sincerely for your suggestion. We conducted transferability testing on targeted attacks [3] (referred to as S²I-SI-TI-DIM) with the hyperparameters set as reported in the original paper. The corresponding results are as follows (some parts have been omitted due to word limit):
> > >
> > > **Complement for Table1**
> > >
> > > |        Src         |    Attack     | →VGG19bn | →Dense121 | →Res152 | →Inc-v3 | →Inc-v4 | →IncRes-v2 |  →ViT   |   AVG   |
> > > |:-----------------:|:-------------:|:--------:|:---------:|:-------:|:-------:|:-------:|:----------:|:-------:|:-------:|
> > > |          Res50         | S²I-SI-TI-DIM |  26.49%  |  36.36%   | 36.49%  | 14.87%  | 12.67%  |   10.02%   |  1.18%  | 19.72%  |
> > >
> > >
> > > |        Src        |    Attack     | →VGG19bn |  →Res50   | →Res152 | →Inc-v3 | →Inc-v4 | →IncRes-v2 |  →ViT   |   AVG   |
> > > |:-----------------:|:-------------:|:--------:|:---------:|:-------:|:-------:|:-------:|:----------:|:-------:|:-------:|
> > > |          Dense121         | S²I-SI-TI-DIM |  25.60%  |  30.33%   | 23.98%  | 13.11%  | 12.04%  |   7.93%    |  1.38%  | 16.33%  |
> > >
> > >
> > > |        Src        |    Attack     |  →Res50  | →Dense121 | →Res152 | →Inc-v3 | →Inc-v4 | →IncRes-v2 |  →ViT   |   AVG   |
> > > |:-----------------:|:-------------:|:--------:|:---------:|:-------:|:-------:|:-------:|:----------:|:-------:|:-------:|
> > > |          VGG19bn         | S²I-SI-TI-DIM |  12.09%  |  14.84%   |  6.49%  |  4.87%  |  5.80%  |   2.93%    |  0.31%  |  6.76%  |
> > >
> > >
> > > |        Src        |    Attack     | →VGG19bn |  →Res50   | →Dense121 | →Inc-v3 | →Inc-v4 | →IncRes-v2 |  →ViT   |   AVG   |
> > > |:-----------------:|:-------------:|:--------:|:---------:|:-------:|:-------:|:-------:|:----------:|:-------:|:-------:|
> > > |          Res152         | S²I-SI-TI-DIM |  19.29%  |  36.64%   | 30.78%  | 14.36%  | 11.42%  |   10.84%   |  1.29%  | 17.80%  |
> > >
> > >
> > > **Complement for Table2**
> > >
> > > | Src                 | Target             |  S²I-SI-TI-DIM  |
> > > |---------------------|----------------------|---------------|
> > > | Res50               | →Inc-v3_adv          | 0.76%         |
> > > |                     | →IncRes-v2_ens       | 0.89%         |
> > > | Dense121            | →Inc-v3_adv          | 0.56%         |
> > > |                     | →IncRes-v2_ens       | 0.84%         |
> > > | VGG19bn             | →Inc-v3_adv          | 0.16%         |
> > > |                     | →IncRes-v2_ens       | 0.22%         |
> > > | Res152              | →Inc-v3_adv          | 0.93%         |
> > > |                     | →IncRes-v2_ens       | 1.02%         |
> > > | Ensemble            | →Inc-v3_adv          | 2.51%         |
> > > |                     | →IncRes-v2_ens       | 2.51%         |
> > >
> > >
> > >
> > > **Complement for Table4**
> > > | Attack          | →VGG19bn | →Dense121 | →Res152  | →IncRes-v2 | →ViT   | AVG    |
> > > |:-----------------:|:----------:|:------------:|:----------:|:--------------:|:---------:|:--------:|
> > > | DTMI-Logit-SU   | 17.82%   | 21.07%     | 21.36%   | 16.38%       | 2.38%   | 15.80% |
> > > __________________________________________________________________________________________________________
> > >
> > > **Response to the question**:
> > > Sincerely appreciate your valuable question, which has greatly contributed to enhancing the clarity of our paper. In Algorithm 3, $N$ and $n$ represent the total number of training epochs and the batch size during training, respectively. These revisions have already been diligently incorporated into the refined version of our original submission.

---

> > > > ### Comment · Reviewer_YEMS · 2023-08-19
> > > > **Response to the reply**
> > > >
> > > > Thanks for provide the experiment about [3].
> > > > For my question, you don't get my point. I wonder the value of $N$ and $n$ in your experiment.

---

> > > > > ### Author Response · Authors · 2023-08-20
> > > > > **Response to the comment**
> > > > >
> > > > > Apologize for the misunderstanding. For the cases where the source model is Res50 or Res152, $N$ is choosen as $300$, while for other cases, $N$ is choosen as $350$. For the cases where the source model is VGG19bn or Ensemble, $n$ is choosen as $20$, whereas for other cases, $n$ is choosen as $25$. We hope this resolves your concerns.

---

> > > > > > ### Comment · Reviewer_YEMS · 2023-08-21
> > > > > > **Response to the reply**
> > > > > >
> > > > > > Why don't adopt the same setting for different source model? Would the experimental results based on several source models be poor if the same setup is used?

---

> > > > > > > ### Author Response · Authors · 2023-08-21
> > > > > > > **Response to the comment**
> > > > > > >
> > > > > > > **Response to the comment**: Sincerely appreciate your comment. Similar to the approach in [1] and [2], we initially set the same number of training epochs, 350 epochs, for all source models. However, we found that for cases where the source model is Res50 or Res152, the test results after 300 and 350 epochs were very close (the results for 350 epochs were shown in the following table, the numbers in parentheses indicate training duration in seconds), the performance improvement brought by the last 50 epochs is less than 5e-3. Following the early-stopping concept and considering practical efficiency, we decided to set $N$ as 300 for Res50 and Res152. But this still guarantees the performance of our method.
> > > > > > > Taking your suggestion into account, we will change the results in our paper, report the results of 350 epochs in all cases.
> > > > > > > The selection of $n$ is determined by the maximum batch size supported by the device.
> > > > > > >
> > > > > > >
> > > > > > > | SRC | Method | →VGG19bn | →Dense121 | →Res152 | →Inc-v3 | →Inc-v4 | →IncRes-v2 | →ViT |  AVG   |
> > > > > > > |:---:|:------:|:--------:|:---------:|:-------:|:-------:|:-------:|:----------:|:----:|:------:|
> > > > > > > | Res50  | ESMA(94721.5) |  81.53%   |  84.46% |  82.04% |  37.82% |  39.60%  | 17.62% | 21.66% | 52.10% |
> > > > > > >
> > > > > > > | SRC | Method | →VGG19bn |  →Res50   | Dense121 | →Inc-v3 | →Inc-v4 | →IncRes-v2 | →ViT |  AVG   |
> > > > > > > |:---:|:------:|:--------:|:---------:|:-------:|:-------:|:-------:|:----------:|:----:|:------:|
> > > > > > > | Res152 | ESMA(110067.2) | 79.27% | 88.64% | 80.60% | 42.02% | 34.73% | 14.91% | 19.18% | 51.33% |
> > > > > > >
> > > > > > >
> > > > > > >
> > > > > > > [1] Muzammal Naseer, Salman H. Khan, Munawar Hayat, Fahad Shahbaz Khan, and Fatih Porikli. On generating transferable targeted perturbations. ICCV, 2021.
> > > > > > > [2] Yang X, Dong Y, Pang T, et al. Boosting transferability of targeted adversarial examples via hierarchical generative networks. ECCV, 2022.

---

> > > > > > > > ### Comment · Reviewer_YEMS · 2023-08-21
> > > > > > > > **Response to the reply**
> > > > > > > >
> > > > > > > > Thanks for your reply. I still have a quetsion: how to calculate training duration for TTP and your proposed method?

---

> > > > > > > > > ### Author Response · Authors · 2023-08-21
> > > > > > > > > **Response to the comment**
> > > > > > > > >
> > > > > > > > > For TTP, we calculate the sum of training time for models across all target classes. For ESMA, we calculate the sum of its training time (including pretrain embedding time) to ensure fairness.

---

> > > > > > > > > > ### Comment · Reviewer_YEMS · 2023-08-21
> > > > > > > > > > **My final comment**
> > > > > > > > > >
> > > > > > > > > > Thanks for your reply. I decide to raise my rating.
> > > > > > > > > > The concerns and questions I have raised may be faced by other readers, and I hope the authors can further polish their papers to make the paper clearer.

---

> > > > > > > > > > > ### Author Response · Authors · 2023-08-21
> > > > > > > > > > > **Response to the comment**
> > > > > > > > > > >
> > > > > > > > > > > Thank you once again for all your questions and suggestions. They are crucial in improving the quality of our article. We will polish our article further and make additions or modifications based on the discussions.

---

### Official Review · Reviewer_YJBk · 2023-06-29

**Soundness:** 3 good
**Presentation:** 3 good
**Contribution:** 3 good
**Rating:** 6
**Confidence:** 4

**Summary:**

This paper presents a hypothesis that pushing adversarial samples towards high-density regions can enhance the transferability of target attacks. The author provides theoretical analysis and experimental validation to support the rationale behind this hypothesis. Based on this hypothesis, the author designs a new generative attack method that improves the adversarial transferability of target attacks.

**Strengths:**

* The idea of pushing ushing adversarial samples towards high-density regions in target attack is well-motivated
* The theoretical analysis presented in this paper is insightful and solid.

**Weaknesses:**

* The lack of persuasive experimental evidence is a major concern.
    1. Insufficient ablation study: The paper introduces multiple modules in the proposed method, but the author fails to include an ablation study. Without such an analysis, it becomes challenging to ascertain whether the proposed method genuinely enhances adversarial transferability. For example, it would be valuable to investigate if the transferability improvement persists when pretrained embeddings are not utilized.
    2. Limited impact of easy example-based adversarial attacks: The experiments presented in Table 1 demonstrate that the adversarial attacks based on easy examples have limited impact on transferability. Additionally, the experimental description in this section lacks sufficient detail. It would be helpful to clarify whether the baseline methods in Table 1 are based on cross-entropy. If they are, I recommend the author redefine the baselines, such as using target anchors generated from randomly selected samples of the target class.
    3. Lack of validation experiments:  Experiments are required to assess whether the adversarial examples generated by the proposed method exhibit higher density/smaller loss and loss gradients compared to those generated by baseline methods. It is essential for the author to include such validation experiments to provide empirical evidence supporting the claims regarding the improved density and reduced loss characteristics of the generated adversarial examples.

* The analysis and proposed solution in the paper exhibit a gap. The conclusion drawn in the paper suggests that samples with lower loss and smaller gradients tend to have stronger transferability. A straightforward approach to address this would be to minimize not only the loss but also the gradient of the loss during the attack. It raises the question of why the author did not employ this direct and simple solution. Instead, the author chose to modify the loss to make the output of adversarial examples align with the target anchor. This approach introduces a gap between the previous analysis and the proposed solution.

* Section 3 lacks clarity in its description. The role of pretrained embedding is not well-defined. I recommend the author to provide a detailed formula or explanation of how pretrained embedding is utilized in the generator.

**Questions:**

*  I suggest the authors conduct ablation studies to verify their claim.
*  Does making the output of adversarial examples match that of the target samples improve the density of adversarial examples?

**Limitations:**

The authors adequately addressed the limitations

---

> ### Author Rebuttal · Authors · 2023-08-08
>
> Sincerely appreciate your valuable suggestions. Here is our sincere response to each of your suggestions (due to word limitations, a portion of the questions you raised has been indicated with ellipsis) :
>
> **Concern1**：The paper introduces multiple modules in the proposed method ...... when pretrained embeddings are not utilized.
>
> **Response to Concern1**: Thank you for your suggestions. In the initial submission, we included some ablation experiments in Appendix C, but indeed did not cover experiments on the performance improvement of pretrained embeddings. Appendix B provides an analysis of the pretrained embeddings. Based on your suggestion, we have incorporated a ablation experiment on pre-trained embeddings in Appendix C. We tested the performance difference of ESMA with and without pre-trained embeddings at different training stages. The results indicate that the model performs noticeably weaker without the pre-trained embedding operation, confirming the effectiveness of this module. We sincerely acknowledge the valuable contribution of your suggestion.
>
> **Concern2**：The experiments presented in Table 1 demonstrate that the adversarial attacks based on easy examples have limited impact on transferability. Additionally, the experimental description in this section lacks sufficient detail. It would be helpful ...... generated from randomly selected samples of the target class.
>
> **Responce to Concern2**: We extend our sincere gratitude for your valuable input. Your insights have the potential to further enhance the quality of our paper. In Table 1, we aimed to offer a straightforward validation of the broad effectiveness of our approach in enhancing target attack transferability. The baseline incorporated therein does indeed entail attacks using standard cross-entropy. Following your suggestion, we have included an additional baseline experiment involving random anchor sample selection for comparison. The performance of the baseline with randomly selected anchors is inferior to the case with sample screening. The comparative ablation experiment on random sample selection and target anchor screening for ESMA has been included in Appendix C of the initial submission.
>
> **Concern3**：Experiments are required to assess whether the adversarial examples generated by the proposed method ...... improved density and reduced loss characteristics of the generated adversarial examples.
>
> **Response to Concern3**: We sincerely appreciate your insightful suggestion, as it holds significant merit. However, directly substantiating the elevation in density presents a formidable challenge. However, directly verifying the increase in density is quite challenging. Our analytical conclusion is that samples with smaller loss and loss gradient norms in the training set often correspond to HSDR, but it does not necessarily imply that samples with smaller loss and loss gradient norms are always within the HSDR. Therefore, conducting this validation experiment is indeed difficult. Your understanding is greatly appreciated.
>
> **Concern4**：The conclusion drawn in the paper suggests that samples with lower loss and smaller gradients ...... This approach introduces a gap between the previous analysis and the proposed solution.
>
> **Response to Concern4**: We are immensely grateful for your insightful inquiry. Specifically, our perspective can be likened to proposing a necessary condition: samples that satisfy this condition within the training set are more likely to reside within the HSDR. However, it does not imply that merely possessing low loss and low loss gradients guarantees a sample's location within such a region. Thus, directly optimizing for loss and loss gradients is insufficient. As a result, we adopt a strategy of aligning by selectively screening such samples as anchor points from the early-stopping model's training set. Your thoughtful consideration is highly appreciated.
>
> **Concern5**：Section 3 lacks clarity in its description. The role of pretrained embedding is not well-defined. I recommend the author to provide a detailed formula or explanation of how pretrained embedding is utilized in the generator.
>
> **Response to Concern5**: We sincerely appreciate your question, as it contributes to further enhancing the clarity of our article. Following your valuable suggestion, we have provided a comprehensive explanation of the specific embedding utilization approach in the appendix of our paper. Allow us to elucidate the details here as well. The trained embedding is denoted as $e_j\in\mathbb R^{d_1}, j\in [K]$, where $e_j\in\mathbb R^{d_1}, j\in [K]$. For the utilized Resblock, let the input and output channels be $C_{in}$ and $C_{out}$ respectively. The fully connected layer $\mathrm{MLP}(\cdot)\in \mathbb R^{d_1}\rightarrow \mathbb R^{C_{out}}$. Firstly, we employ the initial convolution kernel Conv1 to map the raw data from the input $h$ of the current layer, transitioning it from [$C_{in}$, H, W] to [$C_{out}$, H, W]. Then we perform an addition operation between this mapped result and the embedding transformation obtained through the MLP. Formally expressed as:
>
> \begin{equation*}
> h' = \mathrm{Conv1}(h)+\mathrm{MLP}(e_{y})
> \end{equation*}
> where $y$ is the label of the input. Subsequently, we apply a second convolution to this output, followed by a LayerNorm operation, and finally, pass it through an attention operation before adding it to the residual. Formally stated as:
> \begin{equation*}
> \mathrm{output}=\mathrm{Attention}(\mathrm{LN}(\mathrm{Conv2}(h'))+h)
> \end{equation*}
>
> Once again, we sincerely appreciate all your insights and suggestions. Your input has been immensely valuable to us. Regarding the added images and tables, you can find them in the PDF file attached in the author rebuttal. The additional experiments are denoted with blue markings in the tables.

---

> > ### Comment · Reviewer_YJBk · 2023-08-19
> >
> > I am sincerely appreciative of the author's response. While the author has indeed addressed certain concerns, there remain a few issues that have not been sufficiently resolved.
> >
> > 1. Notably, within the author's response, there is an absence of validation to the generated samples being HSDR. It becomes essential to inquire how the enhancement of transferability can be unequivocally attributed to HSDR rather than other contributing factors, given the lack of this validation. In my view, such substantiation is of utmost necessity.
> > 2. Additionally, it is advisable for the author to provide further clarification regarding the relationship between HSDR and the concept of "low loss and low loss gradients." This elucidation would serve to enhance the reader's comprehension of the intricate interplay between these elements.

---

> > > ### Author Response · Authors · 2023-08-20
> > > **Response to the comments**
> > >
> > > Thank you sincerely for your comments. Below are our responses to each of your suggestions:
> > > **Response to comment1**:
> > > Sincerely appreciate your suggestion, which is crucial for improving our article. We would like to clarify that such perturbations do not guarantee that the purturbated samples will be precisely within the HSDR. However, they tend to move towards the HSDR of the target class. To verify whether ESMA can indeed increase the sample density of the original samples, we selected samples from ImageNet that used for adversarial testing and conducted experiments following these steps:
> > >
> > > We calculated the target class sample density, denoted as $\rho_{(y_{\text{tar}}, x_i, r)}$ (here, $r$ is set to 600), for each clean sample $x_i$ in 10 target classes. Then, we averaged the densities with respect to the number of target classes.
> > >
> > > Using ESMA, the original samples were perturbed towards each of the 10 target classes sequentially. For each obtained perturbed sample $x_i^{\text{adv}}$, we calculated the density $\rho_{(y_{\text{tar}}, x_i^{\text{adv}}, r)}$ and averaged it with respect to the number of target classes.
> > >
> > > To normalize the two sets of values into the range of 0-1, we divided them by the maximum value in each set. We then divided the range of 0-1 equally into 10 bins and counted the number of $x_i$ and $x_i^{\text{adv}}$ in each bin. Additionally, we calculated the means of the two sets of the normalized values.
> > >
> > > The final results are presented in the table below:
> > >
> > > **Case for $\epsilon=8/255$**:
> > > | Counts | 0-0.1 | 0.1-0.2 | 0.2-0.3 | 0.3-0.4 | 0.4-0.5 | 0.5-0.6 | 0.6-0.7 | 0.7-0.8 | 0.8-0.9 | 0.9-1.0 |
> > > |:------:|:-----:|:-------:|:-------:|:-------:|:-------:|:-------:|:-------:|:-------:|:-------:|:-------:|
> > > | Clean  |   32  |    35   |    31   |    45   |    39   |    51   |    57   |    51   |    61   |    48   |
> > > | ESMA |   27  |    30   |    32   |    37   |    43   |    49   |    62   |    51   |    70   |    49   |
> > >
> > > **Mean-Density-Clean**: 0.55; **Mean-Density-ESMA**: 0.57
> > >
> > > **Case for $\epsilon=16/255$**:
> > > |   Counts   | 0-0.1 | 0.1-0.2 | 0.2-0.3 | 0.3-0.4 | 0.4-0.5 | 0.5-0.6 | 0.6-0.7 | 0.7-0.8 | 0.8-0.9 | 0.9-1.0 |
> > > |:----------:|:-----:|:-------:|:-------:|:-------:|:-------:|:-------:|:-------:|:-------:|:-------:|:-------:|
> > > |   Clean    |   32  |   35    |   31    |   45    |   39    |   51    |   57    |   51    |   61    |   48    |
> > > | ESMA |   25  |   27    |   36    |   30    |   49    |   50    |   59    |   54    |   71    |   49    |
> > >
> > > **Mean-Density-Clean**: 0.55; **Mean-Density-ESMA**: 0.58
> > >
> > > The clean samples exhibit a significant increase in the first four bins, while the perturbed samples after ESMA show a noticeable decrease in the first four bins and a clear increase in the subsequent bins. These findings support our viewpoint, and this experiment will be reported in our appendix.
> > >
> > > **Response to comment2**:
> > > Sincerely appreciate your suggestion. We will explicitly clarify this relationship in the paper to facilitate readers' understanding.

---

> > > > ### Comment · Reviewer_YJBk · 2023-08-21
> > > >
> > > > Thanks for the author's reply. I decide to raise my score due to the address of my major concern. I hope the authors provide further clarification regarding the relationship between HSDR and the concept of "low loss and low loss gradients" in the draft revision.

---

> > > > > ### Author Response · Authors · 2023-08-21
> > > > > **Reponse to the comment**
> > > > >
> > > > > Once again, we sincerely appreciate your questions and suggestions, which play a vital role in enhancing the quality of our article.

---

### Official Review · Reviewer_tYob · 2023-06-29

**Soundness:** 4 excellent
**Presentation:** 3 good
**Contribution:** 3 good
**Rating:** 7
**Confidence:** 4

**Summary:**

The paper presents both theoretical and experimental evidence showing that incorporating perturbations towards the High-Sample-Density-Regions (HSDR) of the target class can significantly enhance the transferability of targeted attacks. To address the challenges of density estimation in high-dimensional data, a novel method for identifying samples within HSDR is proposed. Leveraging these insights, the authors introduce ESMA, which outperforms existing generative targeted attacks in terms of effectiveness and efficiency.

**Strengths:**

The paper provides theoretical and experimental evidence to support the observation that deep learning models exhibit more consistent outputs in the High-Sample-Density-Regions (HSDR) of each class (Section 2.1). This finding establishes a foundation for the proposed approach. The paper demonstrates through experimental and theoretical analysis that easy samples with low loss in early-stopping models are commonly found in HSDR. Leveraging this insight, the proposed method directly adds perturbations targeting these samples of the target class, resulting in improved targeted adversarial transferability. This approach circumvents the need for computationally expensive density estimation of high-dimensional samples. Finally the paper introduces the Easy Sample Matching Attack (ESMA)  which surpasses state-of-the-art generative attacks such as TTP in terms of targeted transfer success rate. Notably, ESMA only requires one model to perform attacks for all target classes significantly reducing storage space requirements and training time compared to TTP.

**Weaknesses:**

There are some grammar mistakes (in lines 231-238, the sentences in the whole paragraph are all connected with commas, making it difficult to read) and the notion is a little bit confusing (e.g. in Algorithm 2, $a_k$ is in the feature space, but in the equation in line 251, $a_j$ is in the input space).

**Questions:**

 1 According to Algorithm 3,  the target class is randomly sampled from a uniform distribution in the training stage. How does the generator generate a target adversarial example given a specific target class in the reference stage?

2 Is the model employed to select target anchors different from the source model in ESMA? This information is not clear.

**Limitations:**

The writing and the usage of notation need to be enhanced.

---

> ### Author Rebuttal · Authors · 2023-08-08
>
> Thank you very much for highlighting these issues, and we greatly appreciate your thorough examination of our work. Below are our responses to each of these questions and suggestions:
>
> **Concern1**：There are some grammar mistakes (in lines 231-238, the sentences in the whole paragraph are all connected with commas, making it difficult to read) and the notion is a little bit confusing (e.g. in Algorithm 2, $a_k$ is in the feature space, but in the equation in line 251, $a_j$ is in the input space).
>
> **Response to Concern1**: We greatly appreciate your identification of the detailed issues in our paper. Following your suggestions, we have made the following revisions:
>
> **Original**:
> To obtain better embeddings that more effectively represents inter-class information, since the latent space of deep learning models often extracts enough class information, we treat the output features of the local model as a set of a priori embedding, in order to make the generator embedding have a similar structure to such a priori embedding, we refer to the idea of manifold learning, using a strategy similar to the SNE algorithm, let $\mu_j =\frac{1}{\left|\mathcal I_j\right|}\sum_{i\in \mathcal I_j}l_i$, where $l_i=f(x_i)$, and then we design the following manifold matching loss , generator embeddings of various class were pulled to the manifold that output features in to obtain embeddings with a better structure.
>
> **Revised**:
> To obtain better embeddings that more effectively represent inter-class information, since the latent space of deep learning models often extracts sufficient class information, we treat the output features of the local model as a set of priori embeddings. In order to make the generator embedding have a similar structure to such priori embeddings, we draw inspiration from manifold learning and employ a strategy similar to the SNE algorithm. Specifically, we introduce $\mu_j =\frac{1}{\left|\mathcal I_j\right|}\sum_{i\in \mathcal I_j}l_i$, where $l_i=f(x_i)$, and then formulate the following manifold matching loss. This loss encourages the generator embeddings of various classes to be attracted to the manifold representing the output features, thus yielding embeddings with a better structure.
>
> Furthermore, we have corrected the reference from $f(a_j)$ in line 251 to $a_j$.
> And we performed a thorough review of the entire paper and carried out a comprehensive refinement to enhance its overall quality.
>
> **Question1**： According to Algorithm 3, the target class is randomly sampled from a uniform distribution in the training stage. How does the generator generate a target adversarial example given a specific target class in the reference stage?
>
> **Response to Question1**: Sincerely appreciate your inquiry. In the reference stage, adversarial samples are generated directly by designating the target classes, eliminating the need for random selection. Each individual sample undergoes an attack targeting all other specified classes. It's important to note that these target classes can be customized and explicitly specified according to the experiment's requirements.
>
> **Question2**：Is the model employed to select target anchors different from the source model in ESMA? This information is not clear.
>
> **Response to Question2**: We sincerely appreciate your question, as it greatly contributes to the refinement of our article. In reality, the selection of target anchors is conducted based on the source model. Our pre-trained embeddings and target anchor selection solely rely on information from the source model. We consciously opted for this approach to maintain fairness and equity in our study. We will also clarify this in the article.
>
> Thank you once again for pointing out issues and raising questions. Your input has been immensely valuable to us.

---

> > ### Comment · Reviewer_tYob · 2023-08-18
> > **Thanks for the response**
> >
> > The authors have addressed my questions, and I intend to keep my current rating.

---

> > > ### Author Response · Authors · 2023-08-18
> > > **Reponse to the comment**
> > >
> > > Thanks for your reply, we sincerely appreciate all of your valuable feedback and suggestions. They have been tremendously helpful to us.

---

### Official Review · Reviewer_YLC1 · 2023-07-06

**Soundness:** 3 good
**Presentation:** 3 good
**Contribution:** 3 good
**Rating:** 4
**Confidence:** 5

**Summary:**

This paper finds that optimizing perturbations towards easy examples located in HSDR can improve targeted adversarial transferability. The authors provide toy experiments and theoretical proof to illustrate these conclusions. They further utilize easy examples to guide the targeted attack in the GAN-like framework. This framework makes it feasible to perform the multi-target attack.

**Strengths:**

1. Exploring targeted adversarial transferability in terms of High-Sample-Density-Regions is interesting.
2. The proposed framework support multi-target adversarial perturbation generation.
3. The motivation of this paper is clear.


**Weaknesses:**

The experiments are insufficient. They miss previous works, hierarchical generative networks for multi-target attack [A], and iterative attacks [B,C,D, E]. Besides, the authors only conduct experiments on naturally trained models, and neglect to consider adversarially trained models and other defense methods. And the authors should attack more diverse network architectures, such as ViTs.


[A] Yang X, Dong Y, Pang T, et al. Boosting transferability of targeted adversarial examples via hierarchical generative networks[C]//European Conference on Computer Vision. Cham: Springer Nature Switzerland, 2022: 725-742.
[B] Zhao Z, Liu Z, Larson M. On success and simplicity: A second look at transferable targeted attacks[J]. Advances in Neural Information Processing Systems, 2021, 34: 6115-6128.
[C] Qin Z, Fan Y, Liu Y, et al. Boosting the transferability of adversarial attacks with reverse adversarial perturbation[J]. Advances in Neural Information Processing Systems, 2022, 35: 29845-29858.
[D] Wei Z, Chen J, Wu Z, et al. Enhancing the Self-Universality for Transferable Targeted Attacks[C]//Proceedings of the IEEE/CVF Conference on Computer Vision and Pattern Recognition. 2023: 12281-12290.
[E] Byun J, Cho S, Kwon M J, et al. Improving the transferability of targeted adversarial examples through object-based diverse input[C]//Proceedings of the IEEE/CVF Conference on Computer Vision and Pattern Recognition. 2022: 15244-15253.

**Questions:**

Please conduct more experiments as discussed above.

**Limitations:**

None.

---

> ### Author Rebuttal · Authors · 2023-08-08
>
> We sincerely appreciate your thorough review of our paper and the valuable feedback you have provided. Allow us to address your inquiries in the following response:
>
> **Concern1**： They miss previous works, hierarchical generative networks for multi-target attack [A], and iterative attacks [B, C , D, E].
>
> **Response to Concern1**: Sincerely appreciate your insightful feedback. We have incorporated the generative attack algorithms [A], [B], and [C] into our original comparisons, denoted as HGN, Logits, and RAP-LS respectively. As the work presented in reference [D] focuses on transferability in the context of UAP scenarios, and reference [E] employs 3-D transformations to enhance transferability, which significantly alter the original images themselves, they do not directly align with our research scenario. However, for the sake of completeness in discussing related work, these mentioned papers are duly referenced and acknowledged in our article.
>
> **Concern2**：Besides, the authors only conduct experiments on naturally trained models, and neglect to consider adversarially trained models and other defense methods.
>
> **Response to Concern2**: We greatly appreciate your valuable suggestions. As the theoretical analysis and fundamental assumptions of this paper are based on the condition of training data from the same distribution, the previous versions did not encompass adversarial training models or models with defense mechanisms. Inspired by your insightful reminder, we have conducted an evaluation of model transferability under adversarial training conditions, and have included this result in the appendix of the article. We have observed certain performance degradation with the proposed methods, which aligns with our initial assumptions. In our subsequent work, we will specifically address the challenges of transferability in scenarios involving distributional non-consistency. Your input has been instrumental in guiding our research direction.
>
> **Concern3**：And the authors should attack more diverse network architectures, such as ViTs.
>
> **Response to Concern3**: We greatly appreciate your suggestion regarding conducting experiments with additional model architectures. We have taken this suggestion into account and have included the Vision Transformer (Vit) as one of the evaluated models. This addition allowed us to validate the effectiveness of our method on the Vit model, further emphasizing its superiority.
>
> Once again, thanks for all the insights and suggestions. Regarding the included comparative experiments, you can find them highlighted in blue in the PDF attached within the Author Rebuttal.

---

> > ### Comment · Reviewer_YLC1 · 2023-08-15
> > **Response to the rebuttal**
> >
> > Thanks for your rebuttal.
> > For Concern1: [D] is the transferable targeted attack by jointly optimizing the global and local images, and it is not in the context of UAP scenarios.
> > For Concern2: From Table 2 in the appendix, the performance of HGN and TTP is better than the proposed ESMA. The authors attribute this poor performance to the different distribution of adversarial examples. However, adversarially trained models also achieve high accuracy on benign inputs. Besides, the poor performance in attacking adversarially trained models exposes the vulnerability of the proposed method.
> > For Concern3: Thanks for the results.

---

> > > ### Author Response · Authors · 2023-08-16
> > > **Response to the comment**
> > >
> > > **Response to Concern1**: Thank you for your response. [D] utilizes a combination of local and global information to perturb and achieves Self-Universality. It is indeed discussed on the scenario of targeted transferable attack as well. We considered your suggestion and included [D] in the comparison (referred to as DTMI-Logit-SU). The corresponding results are as follows (due to word limit, we omit other reported rows and columns). It is still inferior to our method.
> > >
> > > **Complement for Table1**
> > > |    Src    |    Attack     | →VGG19bn | →Dense121 | →Res152 | →Inc-v3 | →Inc-v4 | →IncRes-v2 |  →ViT  |  AVG   |
> > > |:---------:|:-------------:|:--------:|:---------:|:-------:|:-------:|:------:|:----------:|:------:|:------:|
> > > | Res50  | DTMI-Logit-SU | 53.36%   | 78.91%    | 79.00%  | 18.56%  | 13.36% |   8.71%    | 3.32%  | 36.46% |
> > >
> > >
> > > |    Src    |     Attack     | →VGG19bn |  →Res50   | →Res152 | →Inc-v3 | →Inc-v4 | →IncRes-v2 |  →ViT   |   AVG   |
> > > |:---------:|:-------------:|:--------:|:---------:|:-------:|:-------:|:------:|:----------:|:------:|:------:|
> > > | Dense121  | DTMI-Logit-SU | 33.38%    | 44.93%    | 34.16%  | 13.44%  | 12.36%  | 7.75%      | 3.62%   | 21.38% |
> > >
> > > |        Src        |    Attack     |  →Res50  | →Dense121 | →Res152 | →Inc-v3 | →Inc-v4 | →IncRes-v2 |  →ViT   |   AVG   |
> > > |:---------:|:-------------:|:--------:|:---------:|:-------:|:-------:|:------:|:----------:|:------:|:------:|
> > > | VGG19bn   | DTMI-Logit-SU | 23.47%    | 31.47%    | 12.60%  | 6.13%   | 7.31%   | 2.07%      | 1.11%   | 12.02% |
> > >
> > > |        Src        |    Attack     | →VGG19bn |  →Res50   | →Dense121 | →Inc-v3 | →Inc-v4 | →IncRes-v2 |  →ViT   |   AVG   |
> > > |:---------:|:-------------:|:--------:|:---------:|:-------:|:-------:|:------:|:----------:|:------:|:------:|
> > > | Res152    | DTMI-Logit-SU | 37.76%    | 77.47%    | 62.91%  | 16.02%  | 11.51%  | 8.58%      | 3.02%   | 31.04% |
> > >
> > > **Complement for Table2**
> > >
> > > | Src                 | Target               | DTMI-Logit-SU |
> > > |---------------------|----------------------|---------------|
> > > | Res50               | →Inc-v3_adv          | 0.27%         |
> > > |                     | →IncRes-v2_ens       | 0.22%         |
> > > | Dense121            | →Inc-v3_adv          | 0.20%         |
> > > |                     | →IncRes-v2_ens       | 0.20%         |
> > > | VGG19bn             | →Inc-v3_adv          | 0.13%         |
> > > |                     | →IncRes-v2_ens       | 0.07%         |
> > > | Res152              | →Inc-v3_adv          | 0.26%         |
> > > |                     | →IncRes-v2_ens       | 0.31%         |
> > > | Ensemble            | →Inc-v3_adv          | 0.53%         |
> > > |                     | →IncRes-v2_ens       | 0.24%         |
> > >
> > >
> > >
> > > **Complement for Table4**
> > > | Attack          | →VGG19bn | →Dense121 | →Res152  | →IncRes-v2 | →ViT   | AVG    |
> > > |:-----------------:|:----------:|:------------:|:----------:|:--------------:|:---------:|:--------:|
> > > | DTMI-Logit-SU   | 18.11%   | 30.16%     | 18.73%   | 38.64%       | 2.64%   | 21.71% |
> > > __________________________________________________________________________________________________________
> > >
> > > **Response to Concern2**: Sincerely appreciate your valuable suggestions and insightful input. The main research logic of this paper is to explore the validation of the proposed sample selection method in improving the transferability of adversarial attacks, under the fundamental assumption of the same training set distribution. We provide relevant theoretical proofs and analyses to inspire further research and new perspectives. In order to validate the transfer effectiveness of the proposed algorithms, we conducted main experiments and ablation studies.
> > > In response to your comment, we have also performed relevant verification and analysis considering the presence of distribution shift (Table 2). However, it is important to clarify that these findings are not directly related to the viewpoints, conclusions, and statements of our study, as we have already indicated in our response.
> > > We sincerely appreciate your enlightening input, as it holds significant meaning and can be pursued for targeted discussions in future work. For instance, one potential avenue for further research could focus on the problem of adversarial transfer on models trained on datasets with distribution shift.

---

### Author Rebuttal · Authors · 2023-08-08

We sincerely appreciate the valuable insights and suggestions provided by the esteemed reviewers during the review process. We extend our gratitude for the time and effort you dedicated to scrutinize our paper. Your reviews are of paramount importance to our research endeavor, and your insightful recommendations and feedback have played a crucial role in further refining and enhancing the quality of our paper. For each reviewer's comments, we have addressed each point individually. The following outlines the modifications we have made based on the reviewers' feedback:

1) We have restructured the references. In order to enhance the comprehensiveness of the literature review and to capture readers' interest, we have incorporated the following six relevant works in the field of adversarial attacks into the references section. Additionally, we have expanded the literature review portion in the introduction.

>[1]Yang X, Dong Y, Pang T, et al. Boosting transferability of targeted adversarial examples via hierarchical generative networks[C]//European Conference on Computer Vision. Cham: Springer Nature Switzerland, 2022: 725-742.
>[2] Zhao Z, Liu Z, Larson M. On success and simplicity: A second look at transferable targeted attacks[J]. Advances in Neural Information Processing Systems, 2021, 34: 6115-6128.
>[3] Qin Z, Fan Y, Liu Y, et al. Boosting the transferability of adversarial attacks with reverse adversarial perturbation[J]. Advances in Neural Information Processing Systems, 2022, 35: 29845-29858.
>[4] Zhang et al. Staircase sign method for boosting adversarial attacks. arXiv:2104.09722.
>[5] Wei Z, Chen J, Wu Z, et al. Enhancing the Self-Universality for Transferable Targeted Attacks[C]//Proceedings of the IEEE/CVF
 Conference on Computer Vision and Pattern Recognition. 2023: 12281-12290.
>[6] Byun J, Cho S, Kwon M J, et al. Improving the transferability of targeted adversarial examples through object-based diverse input[C]//Proceedings of the IEEE/CVF Conference on Computer Vision and Pattern Recognition. 2022: 15244-15253.


2) Building upon the existing comparative experiments, we have expanded our evaluation to include a comparison with the methods proposed in the works [1], [2], [3], and [4], denoted as HGN, Logit, RAP-LS, and SI-FGS$^2$M respectively. Additionally, we have introduced the Vision Transformer (ViT) as another tested model. Furthermore, we have included a comparison involving an ensemble model as the source model, which is presented in the appendix. The experimental results consistently affirm the superior transferability of our approach even amidst these additional comparisons.


3) Due to the theoretical analysis and foundational assumptions of this paper being built upon the premise of training data from the same distribution, we conducted additional transfer attack tests on adversarially trained models to explore the transferability of models trained on different distribution data. The experimental results indicate a certain degree of performance degradation in the proposed method, aligning with our initial assumptions. In subsequent work, we will concentrate on addressing the challenges posed by the perturbation transferability in scenarios with different distributions.


4) We have introduced additional ablation experiments comparing models with and without pre-trained embeddings. The experimental results validate the effectiveness of the proposed pre-training operation. Furthermore, in Appendix B, we have included a formulaic representation of the methodology for utilizing pre-trained embeddings to provide readers with greater clarity.


5) We have thoroughly reviewed the grammar, symbols, and formulas within the paper. We have made revisions to address the specific points highlighted by the reviewers. And we have subsequently polished the entire article. This was done to enhance the overall clarity of the content, enabling readers to better comprehend the paper.


All additional experimental tables and figures have been documented in the attached PDF file. For the newly added results, we have utilized blue font for differentiation. Once again, we express our sincere gratitude for the reviewers' patient reading and highly valuable suggestions.

---

### Decision · Program_Chairs · 2023-09-21

**Decision:**

Accept (poster)

**Comment:**

This paper studies the adversarial robustness of neural networks. Specifically, the authors aim to improve the targeted adversarial transferability, where adversarial examples for one "source" white-box model are used to fool another "target" model in a black-box fashion. The main insight is that optimizing perturbations towards easy examples located in the high sample density regions (HSDRs) improves targeted adversarial transferability. The paper contains both a theoretical analysis of why this is effective, as well as experiments on several source and target networks, including ResNet and Inception. The paper also contains toy examples (e.g., Gaussian mixture models) to demonstrate the approach.

Reviewers found many positive aspects to the paper. They found the general approach of using HSDRs to be compelling, intuitive, and effective. They also appreciated that the method can support multi-target perturbations. The reviewers had many questions about the experiments, but overall, they found that the new proposed algorithm (ESMA) to be quite effective and better than several prior works / baselines. The reviewers also found the method to be fairly practical and usable, given that the authors specifically explain how to implement it for high dimensional datasets (i.e., ImageNet-1k).

On the negative side, the reviewers had many questions about ablation studies and related work. For example, there is a question about whether the proposed method works without using pre-trained embeddings (this seems to be a key assumption). During the discussion, the authors added a variety of new experiments to (at least partially) address these concerns. Some other concerns were around tying claims about the density to the adversarial attack effectiveness, as well as trying out model ensembles. The authors have added new experiments to help illuminate these topics. For example, the authors have added new experiments to answer: "how the enhancement of transferability can be unequivocally attributed to HSDR rather than other contributing factors?"

In summary, the discussion was extremely fruitful, and assuming all the new experiments end up in the final version of the paper, the authors' claims seem to be very well supported. The original results already demonstrated significant improvements. Therefore, I recommend acceptance of the paper.

I strongly encourage the authors to incorporate all of the additions that they mention during the discussion in the final version of the paper, both in the appendix and with some references in the main paper to improve easy identification of answers for the main reviewer concerns.